# A pediatric brain tumor atlas of genes deregulated by somatic genomic rearrangement

Yiqun Zhang[1,8], Fengju Chen[1,8], Lawrence A. Donehower[2,3], Michael E. Scheurer[1,4,5] & Chad J. Creighton [1,3,6,7✉]

The global impact of somatic structural variants (SSVs) on gene expression in pediatric brain tumors has not been thoroughly characterised. Here, using whole-genome and RNA sequencing from 854 tumors of more than 30 different types from the Children's Brain Tumor Tissue Consortium, we report the altered expression of hundreds of genes in association with the presence of nearby SSV breakpoints. SSV-mediated expression changes involve gene fusions, altered cis-regulation, or gene disruption. SSVs considerably extend the numbers of patients with tumors somatically altered for critical pathways, including receptor tyrosine kinases (*KRAS*, *MET*, *EGFR*, *NF1*), Rb pathway (*CDK4*), *TERT*, MYC family (*MYC*, *MYCN*, *MYB*), and HIPPO (*NF2*). Compared to initial tumors, progressive or recurrent tumors involve a distinct set of SSV-gene associations. High overall SSV burden associates with *TP53* mutations, histone H3.3 gene *H3F3C* mutations, and the transcription of DNA damage response genes. Compared to adult cancers, pediatric brain tumors would involve a different set of genes with SSV-altered cis-regulation. Our comprehensive and pan-histology genomic analyses reveal SSVs to play a major role in shaping the transcriptome of pediatric brain tumors.

[1] Dan L. Duncan Comprehensive Cancer Center Division of Biostatistics, Baylor College of Medicine, Houston, TX, USA. [2] Department of Molecular Virology and Microbiology, Baylor College of Medicine, Houston, TX, USA. [3] Human Genome Sequencing Center, Baylor College of Medicine, Houston, TX, USA. [4] Department of Pediatrics, Baylor College of Medicine, Houston, TX, USA. [5] Texas Children's Cancer Center, Texas Children's Hospital, Houston, TX, USA. [6] Department of Bioinformatics and Computational Biology, The University of Texas MD Anderson Cancer Center, Houston, TX, USA. [7] Department of Medicine, Baylor College of Medicine, Houston, TX, USA. [8] These authors contributed equally: Yiqun Zhang, Fengju Chen. ✉email: creighto@bcm.edu

Somatic genomic rearrangements in cancer have a widespread impact on the transcriptome[1–5]. All classes of somatic structural variants (SSVs)—including tandem duplications, insertions, deletions, inversions, and translocations[6]—can potentially alter the regulation of specific genes through several possible mechanisms, including gene fusion, promoter element disruption, enhancer hijacking, disruption of topologically associated domains (TADs), and altered DNA methylation[4]. Also, cancers harboring a high overall structural variation burden may exhibit an altered molecular profile reflective of extensive DNA damage[4]. To comprehensively study SSVs and other noncoding genomic alterations in the setting of adult human cancers, the Pan-cancer Analysis of Whole Genomes (PCAWG) consortium recently aggregated whole-genome sequencing (WGS) data from over 2000 cancers across 38 tumor types. These PCAWG data are taken from the International Cancer Genomics Consortium (ICGC) and The Cancer Genome Atlas (TCGA) Consortium[7]. Over 20 studies carried out in conjunction with the PCAWG consortium explored various aspects of noncoding genomic alteration in cancer. Among these studies of the PCAWG sample cohort, we reported that hundreds of genes showed recurrently altered expression in association with the nearby presence of an SSV breakpoint[3]. Genes recurrently upregulated in association with SSVs included known oncogenes—such as TERT, MDM2, CDK4, ERBB2, CD274, PDCD1LG2, and IGF2—while tumor suppressor genes—such as PTEN, STK11, TP53, and RB1—were frequently disrupted by SSV breakpoints falling within the gene body. However, PCAWG expression data did not include any pediatric brain tumor cases, representing a knowledge gap as to how SSVs may alter expression within this patient population.

The Children's Brain Tumor Tissue Consortium (CBTTC) (https://cbttc.org) is an international, collaborative, and multi-institutional research program dedicated to studying childhood brain tumors[8]. CBTTC efforts include providing open access genomics data on pediatric brain tumors to the research community to more fully elucidate these tumors' molecular landscape. Previous studies of other patient cohorts have carried out WGS for specific types of pediatric brain cancer[9–15]. However, the CBTTC data are unique in involving a large set of samples—854—with both WGS and RNA sequencing (RNA-seq) data and spanning multiple histologic types, comparable to the 1220 PCAWG WGS samples with RNA-seq (all but 24 of which were over the age of 17)[3,16]. No CBTTC tumors were included as part of the PCAWG datasets. Using CBTTC genomic data to define new gene targets altered consistently by somatic structural variation would have implications for personalized and precision medicine approaches. Pediatric brain cancer—the most common type of solid tumors in children—is a heterogeneous disease representing numerous distinct histologic types, involving different sets of driver mutations and different responses to therapy[17]. CBTTC data offer pan-cancer analysis opportunities to identify patterns cutting across histologic types of pediatric brain cancer[7,18], in addition to genomic analyses within individual tumor types. WGS analysis approaches, as demonstrated in adult cancers by PCAWG consortium and others, remain to be applied systematically to pediatric cancers, as different genomic loci and associated genes are likely to be targeted in the pediatric tumor setting.

This present study utilizes the CBTTC datasets to analyze high coverage WGS data from 854 pediatric brain tumor samples and 759 individuals. Integrating SSV calls with gene expression data, we observe a widespread impact of SSVs on the regulation of genes in the vicinity of SSV breakpoints, independent of copy-number alterations (CNAs), involving key oncogenes and tumor suppressor genes. In another line of investigation, we examine gene expression and DNA mutation features correlated with the overall SSV tumor burden, independently of where the breakpoints fall in proximity to genes. Here, a high SSV burden is associated with TP53 mutations, histone H3.3 gene H3F3C mutations, and increased expression of DNA damage response genes.

## Results

**SSVs in CBTTC cohort.** Our study focused on 854 pediatric brain tumor samples from the CBTTC, representing 759 patients, for which both WGS and RNA-seq data were available (Supplementary Data 1). Tumor samples in CBTTC spanned at least 33 different tumor types based on histology, the most highly represented of which (13 or more tumors for each) included: low-grade glioma/astrocytoma (PLGG, n = 239 tumors), medulloblastoma (MBL, n = 104), ependymoma (EPMT, n = 79), high-grade glioma/astrocytoma (PHGG, n = 76), ganglioglioma (GNG, n = 48), craniopharyngioma (CRANIO, n = 36), atypical teratoid rhabdoid tumor (ATRT, n = 30), meningioma (MNG, n = 30), dysembryoplastic neuroepithelial tumor (DNT, n = 24), schwannoma (SCHW, n = 16), neurofibroma/plexiform (NFIB, n = 16), choroid plexus papilloma (CPP, n = 14), and supratentorial or spinal cord primitive neuroectodermal tumor (PNET, n = 13). Some tumor types represented in CBTTC—including Ewing's sarcoma (ES, n = 6), germinoma (GMN, n = 4), Langerhans cell histiocytosis (LCH, n = 4), malignant peripheral nerve sheath tumor (MPNST, n = 3), and neuroblastoma (NBL, n = 2)—originate from cell types not specific to the brain, even if the CBTTC tumors were obtained from the brain region. CNAs were examined in these CBTTC tumors, with overall CNA patterns markedly differing from those of adult brain tumors from TCGA (Supplementary Fig. 1a), and with a sizable proportion of CBTTC tumors appearing relatively quiet at the CNA level.

By WGS analysis, a median of 47 SSVs was found per tumor (with standard deviation, or SD, of 167.2). On average, the numbers of SSVs detected varied widely according to tumor type (Fig. 1a). Hemangioblastoma (HMBL), LCH, ES, SCHW, and DNT tumors tended to have the fewest SSVs (average of 37.7 SSVs for these tumor types). MPNST, PHGG, Diffuse intrinsic pontine glioma (DIPG), NBL, and sarcoma (SARCNOS) tumors tended to have the most SSVs (average of 230.8 SSVs). In line with previous observations in adult cancers[3,19], here, genomic rearrangements could be associated with widespread CNA patterns in pediatric brain tumors (Fig. 1b). When we considered the set of all SSV-gene associations involving an SSV breakpoint falling within 1 Mb of gene start site for a given tumor, we found these associations to be highly enriched for gene-level amplification or deletion, though more so for the former. While a significant proportion of SSVs associated with gene amplification involved tandem duplication SSVs as might be expected, all classes of SSV were involved with altered CNA patterns (Supplementary Fig. 1b). The intra-chromosomal, non-translocation SSVs associated with CNA showed enrichment for SSVs of larger DNA sizes (>100 kb, Supplementary Fig. 1c).

A subset of CBTTC tumors represented multiple tumors taken from the same patient, involving 170 tumor samples from 75 patients. These 170 tumors would include samples taken from a patient at different times, e.g., samples taken from an initial tumor and later from a progressive or recurrent tumor, or samples representing a second malignancy (Supplementary Data 1). In other cases (involving 13 patients), multiple initial tumors were profiled, often taken from different anatomic sites, as a patient may present with multiple initial tumors upon diagnosis. We found that different tumors from the same patient tended to demonstrate extensive molecular heterogeneity among

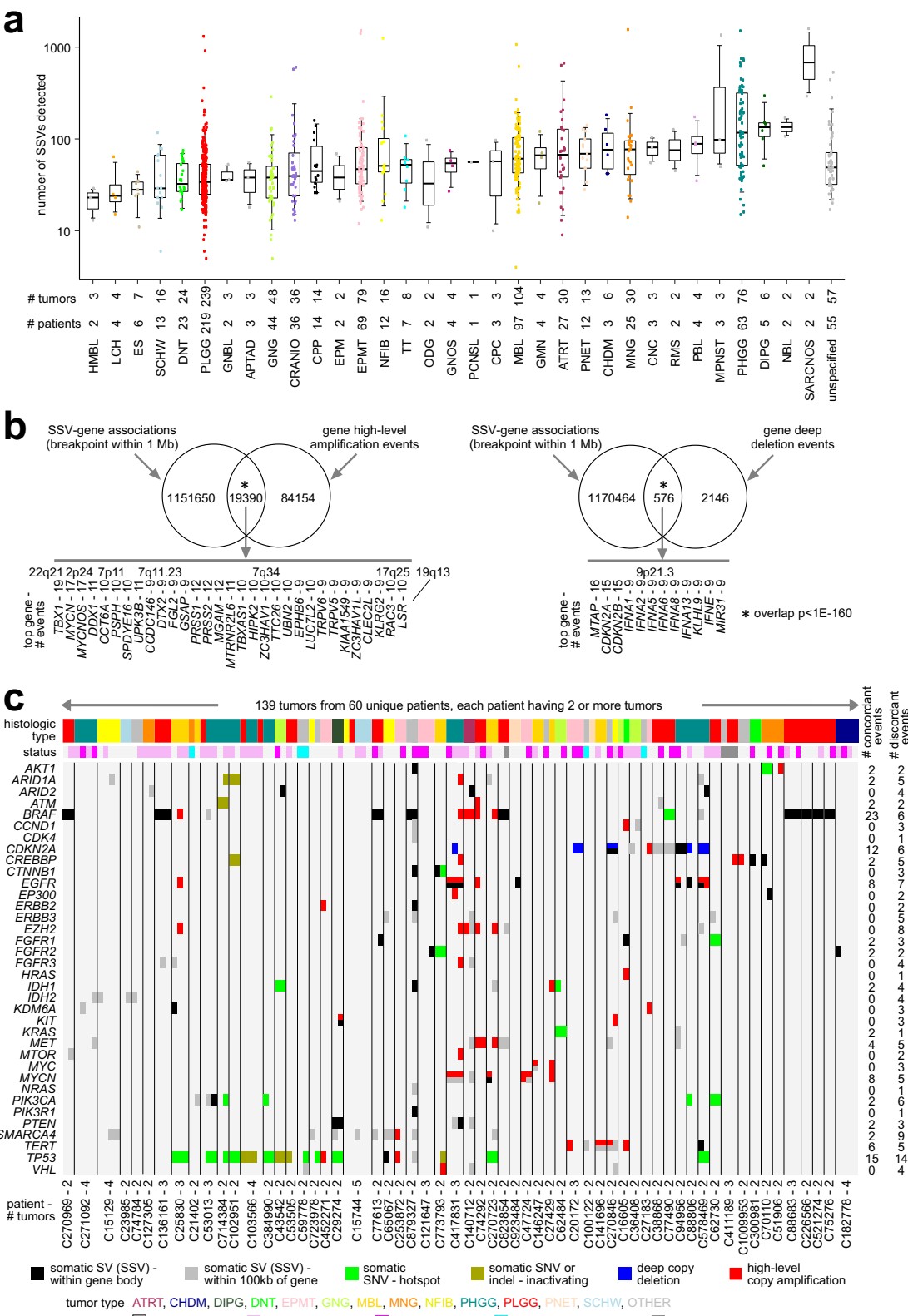

them. Taking an example set of 36 cancer-associated genes[20,21], we examined somatic events, including SSV breakpoints occurring within 100 kb of any genes, across 139 tumors from 60 patients for which multiple tumors were profiled and for which at least one of the 36 genes had breakpoint associations (Fig. 1c). SSV patterns—together with patterns of CNA, insertion/deletion

of nucleotide bases (indels), and Single Nucleotide Variants (SNVs)—revealed both concordant and discordant patterns among tumors from the same patient. Furthermore, in a global analysis of SSV breakpoint patterns across the 75 patients (Supplementary Fig. 1d), inter-profile correlations between different tumors from the same patient ranged from very high

**Fig. 1 SSVs detected across diverse pediatric brain tumor types from the CBTTC. a** By brain tumor type, box plot of numbers of SSVs detected for each CBTTC tumor sample, representing 854 tumors and 759 patients in total. Box plot represents 5% (lower whisker), 25% (lower box), 50% (median), 75% (upper box), and 95% (upper whisker). See Methods regarding histology-based tumor type abbreviations. **b** Association of CNAs with genomic rearrangement. Based on the set of all SSV-gene associations involving an SSV breakpoint falling within 1 Mb of gene start site for a given tumor (taken from all gene X tumor sample pairings), Venn diagrams represent significant enrichment of these SSV-gene associations both for high-level gene amplification events (left) and for deep gene deletion events (right). For each of the overlapping results sets, lists provide the most frequently affected genes and associated cytoband regions. *P* values by chi-squared test. **c** Evidence of inter-tumoral heterogeneity within patients, by SSV breakpoint patterns surveyed across multiple tumors from the same patient. Tumor status color bar denotes initial tumor, progressive, recurrence, or second malignancy. Based on a set of 36 cancer-associated genes (defined using the literature[20,21]), somatic events are represented across 139 tumors from 60 patients for which CBTTC profiled multiple tumors (the 139 tumors involve at least one gene with breakpoint association). Black or gray represents SSV breakpoint in proximity to the gene (within the gene body or 100 kb of the gene, respectively). Green or gold represent somatic SNV/indel (respectively, either missense SNV within hotspot residue[49] or inactivating mutation by indel/nonsense/nonstop). Red or blue represents high-level amplification or deep deletion, respectively. As tabulated on the right, concordant events are somatic events detected in all tumors from the same patient; discordant events are detected in only some but not all tumors from the same patient. Tumor type color scheme is from part a.

to very low, low to the degree of suggesting little relationship between the tumors. These results lent support to our treating each tumor as a separate disease entity in the downstream analyses. In other words, given multiple tumors from the same patient (e.g., involving multiple initial tumors, or initial compared to recurrent or progressive tumors), we considered each tumor sample separately from the others in our analyses.

**Global impact of SSVs on gene expression patterns**. Using integration approaches between SSVs and gene expression, previously demonstrated in adult cancers[4,19,22], we assessed gene-level associations between expression and nearby SSV breakpoints across the CBTTC tumor cohort. In principle, SSVs with breakpoints nearby a gene could lead to altered cis-regulation, e.g., by enhancer hijacking or altered DNA methylation[4], and SSV breakpoints within a gene could result in gene disruption or a gene fusion[19] (Fig. 2a). For each gene with expression data, we assessed the pattern of nearby SSV breakpoints within a given region window (e.g., 100 kb upstream of the gene). From CBTTC data, we assembled a data matrix of breakpoint patterns for all 18786 unique named genes and 854 tumors. We then assessed the association between expression and SSV breakpoint pattern for each gene by linear models correcting for tumor type and gene-level CNA. As intended, our analytical approach does not assume the specific mechanism of altered expression, as there may be multiple mechanisms involved for any given gene across different tumors. In principle, SSVs of any class or size may lead to altered cis-regulation, depending on the specifics involving any nearby rearrangements and the gene regulatory landscape.

Hundreds of genes showed altered gene expression in relation to nearby SSV breakpoints, including breakpoints located either downstream or upstream of genes and breakpoints occurring in the gene body (Fig. 2b and Supplementary Data 2). Incorporating statistical corrections for gene-level CNA decreased the overall numbers of significant genes, reflecting the above associations of SSV breakpoints with copy gain. Many more genes showed positive correlations with SSV breakpoints (i.e., expression was higher when SSV breakpoint was present) than negative correlations. When considering a 1 Mb region window upstream or downstream of each gene (whereby the model weighted the relative gene distances of the breakpoints[4], giving the most weight to breakpoints closest to the gene), 324 genes showed positive correlations with SSV breakpoints, and 51 genes showed negative correlations (FDR < 10%[23] by relative distance metric method[4], with corrections for tumor type and CNA). Genes positively correlated with SSV breakpoints included many known oncogenes, while genes negatively correlated included many known tumor suppressor genes (Fig. 2c), as discussed further below. Out of the 854 CBTTC tumors, 811 (95%) involved overexpression

(SD > 0.4) of a gene from our top set of 324 showing global positive SSV correlations (FDR < 10%), coupled with an SSV breakpoint within 1 Mb of the gene. The above indicates that while individual SSV events involving a particular gene may be relatively infrequent among tumors, the cumulative effect of this phenomenon across many genes would involve the vast majority of tumors.

In addition to results using statistical modeling and the Storey and Tibshirani FDR estimation method[23], permutation testing results reflected widespread significant patterns associating SSVs with expression (Supplementary Fig. 2a, b). By permuting the 854 SSV breakpoint profiles within tumor types 1000 times, the 324 genes with Storey and Tibshirani FDR < 10% (using the 1 Mb window) would have an estimated FDR of 33% by permutation method (Supplementary Data 2), indicating that on the order of 67% of the genes would represent true positives. However, the number of tumor samples in the dataset limited permutation testing power, as some permutations would not be too far removed from the actual dataset. Permutation testing results using a large simulated dataset of over 4000 tumors yielded even fewer estimated false positives from the 854-tumor permutation results (Supplementary Fig. 2a), with ~74% of the 324 genes presumably representing true positives (Supplementary Data 2). With very large sample sizes, datasets would be expected to yield permutation testing results much closer to the Storey and Tibshirani FDR estimates. In our downstream analyses below, we focused primarily on known cancer genes (including known fusions) and global associations involving the 324 genes.

The set of genes having significant associations between expression and nearby SSV breakpoint, as identified in the CBTTC cohort, overlapped significantly with the genes arising from similar analyses in adult cancers from combined TCGA-ICGC cohorts[4] (Fig. 2d). Of the 324 genes with significant positive correlations (using the 1 Mb window), 29 were significant by the same approach as applied to TCGA-ICGC pan-cancer dataset (overlap *p* < 1E-5, one-sided Fisher's exact test), including *CDK4*, *EGFR*, *FOXR1*, *MYC*, and *TERT*. At the same time, this overlap represented just a fraction of the significant CBTTC genes, reflecting overall differences between pediatric brain cancers and adult cancers of various types. Similarly, we evaluated SSV-gene associations in TCGA glioma dataset (*n* = 107 GBM/ LGG cases), where 19 of the 324 CBTTC genes overlapped with the significant TCGA glioma genes (Fig. 2d). However, the vast majority of significant genes were exclusive to either the pediatric brain or adult brain settings (Fig. 2e). Genes significant for the CBTTC dataset but not for TCGA glioma dataset included important oncogenes such as *MYB*, *FOXR1*, *TERT*, *MET*, *MYC*, and *MYCN*. For the 13 CBTTC pediatric brain tumor types with the most tumors (13 or more tumors), we evaluated significant

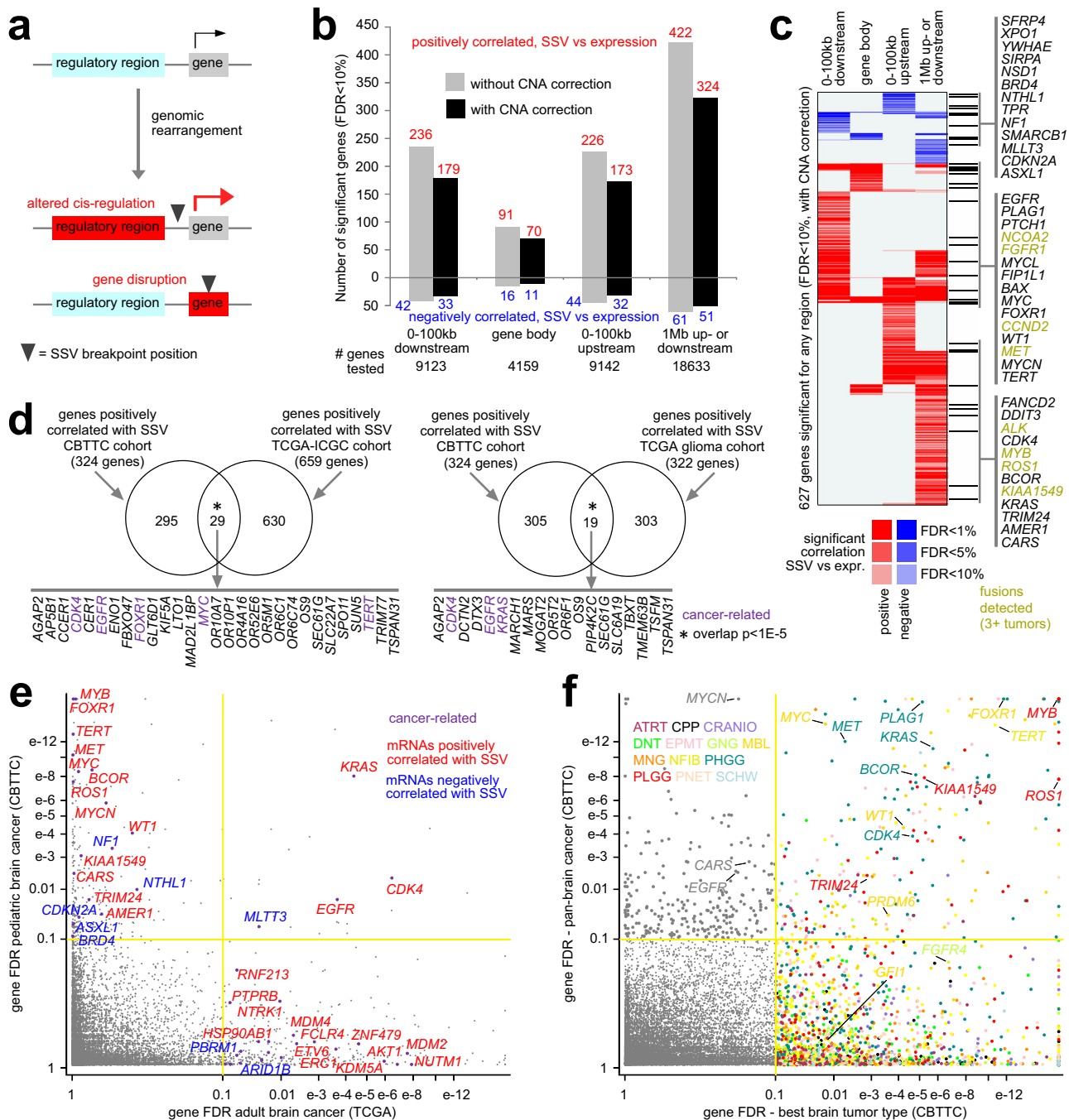

**Fig. 2 Genes with altered expression associated with nearby SSV breakpoints. a** Schematic of the phenomenon of interest. SSV breakpoints outside of genes may result in altered cis-regulation. SSV breakpoints within genes may involve gene disruption or fusions. **b** For each of the indicated genomic region windows examined, numbers of significant genes (FDR < 10%) showing a correlation between expression and associated SSV event. Numbers above and below the zero point of the y-axis denote positively and negatively correlated genes, respectively. Linear regression models evaluated significant associations when correcting for tumor type (gray) and for both tumor type and gene-level CNA (black). For the 1 Mb region window, the model weights the relative gene distances of the breakpoints[4]. **c** Heat map of significance patterns for 627 genes significant for any region window (FDR < 10%, correcting for both tumor type and CNA). Red denotes significant positive correlation; blue, significant negative correlation. Genes listed are cancer-associated[20]. **d** Venn diagrams representing the overlaps between the genes positively correlated (FDR < 10%) with SSV breakpoint in CTBBC pediatric brain cohort and the genes positively correlated (FDR < 10%) with SSV breakpoint in either TCGA-ICGC cohort (left, n = 2334 cases) or TCGA glioma cohort (right, n = 107 GBM/LGG cases). P values by one-sided Fisher's exact test. **e** Significance of genes in TCGA glioma cohort (107 patients, x-axis), as compared to their significance in the CBTTC cohort (854 tumors, y-axis). Genes in the upper left quadrant reached significance only in the present study. **f** X-axis indicates the FDR in the most significant of 13 pediatric brain tumor types analyzed separately, and y-axis indicates the FDR when the 854 CBTTC tumors are analyzed as a combined cohort. Genes in the upper left quadrant reached significance only in the pan-cancer analysis. Genes in the lower right quadrant reached significance only in one or more single-type analyses. Color of data points represents the most significant tumor type. For **d**–**f**, significant genes are defined by 1 Mb region window, correcting for tumor type and CNA. For **d**, **e**, "cancer-related" is by COSMIC[20].

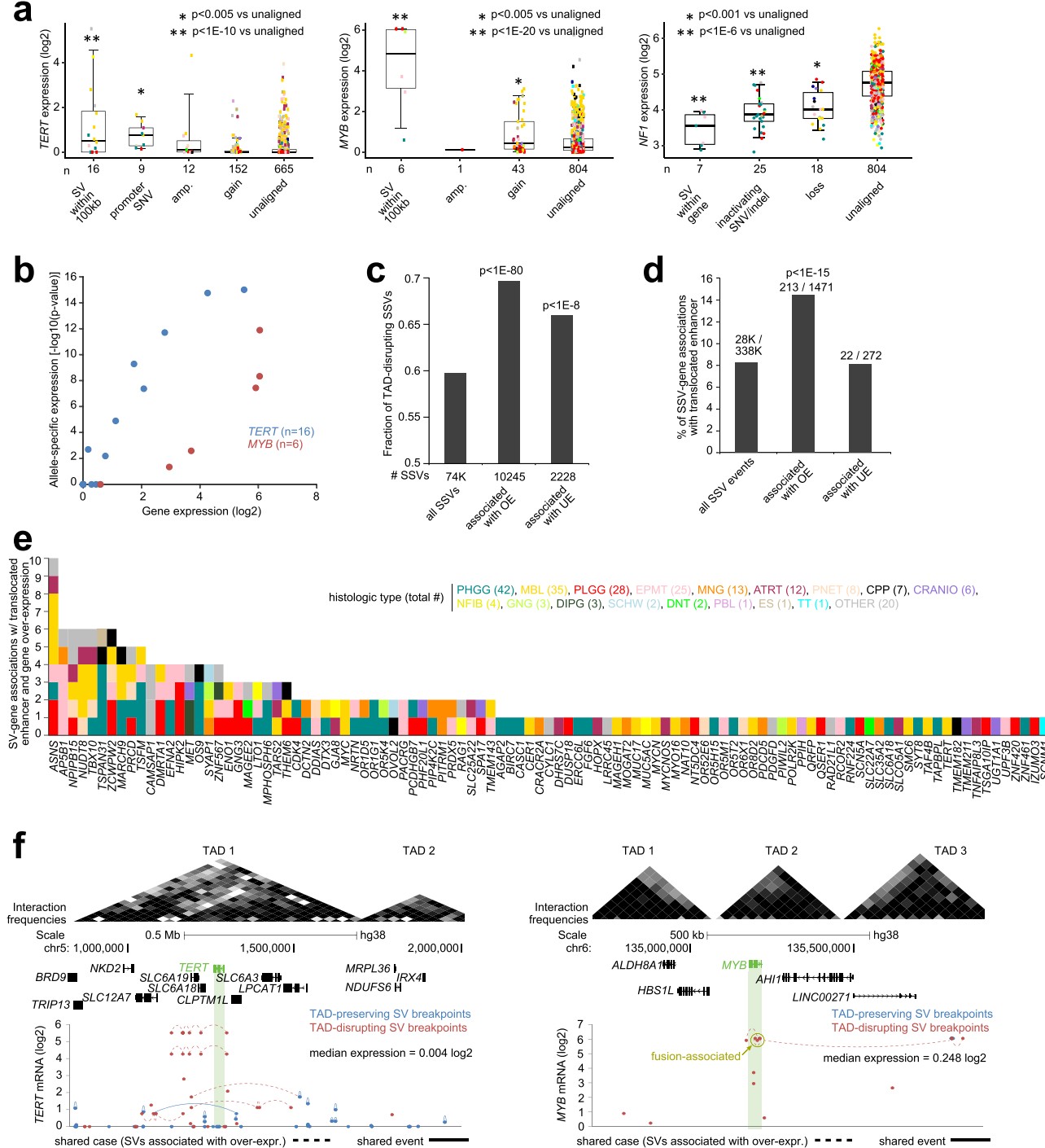

SSV-gene associations separately by tumor type. For most individual tumor types, over 100 significant genes were found (Supplementary Data 3 and Supplementary Fig. 2c). Notably, many genes significant in the analysis of individual tumor types did not reach significance when analyzing the combined pan-CBTTC set (Fig. 2f), analogous to results of adult pan-cancer studies surveying significantly mutated genes[24] or significant SSV associations[4]. SSV-gene associations specific to MBL included *PRDM6* and *GFI1*[10].

A fraction of the positive correlations appeared to reflect SSV-mediated disruption of TAD or enhancer hijacking. For particular genes of interest, including oncogenes *TERT* and *MYB* and tumor suppressor gene *NF1*, the relative expression changes in tumors harboring an SSV breakpoint were more dramatic as compared to

tumors with CNA (Fig. 3a). SSV-associated upregulation of *TERT* and *MYB* appeared allele-specific (Fig. 3b). Using data on TAD coordinates in human cells[25,26], we categorized all SVs in the CBTTC dataset by those that were TAD disrupting (i.e., the breakpoints span two different TADs) versus those that were non-disrupting (i.e., both breakpoints fell within the same TAD). For SSVs associated with gene overexpression, we observed an enrichment for TAD-disrupting SSVs (Fig. 3c, $p < 1E-80$, chi-squared test), consistent with previous observations in adult tumors[19]. Similarly, SSV breakpoints involving gene overexpression were enriched ($p < 1E-15$, chi-squared test) for putative enhancer translocation events, with the rearrangement bringing an enhancer within 500 kb of the gene (Fig. 3d), involving 103 overexpressed genes and 140 tumors (Fig. 3e and Supplementary

**Fig. 3 SSVs associated with disruption of TADs and translocated enhancers. a** Box plots of expression for *TERT*, *MYB*, and *NF1* by alteration class ("amp." or high-level gene amplification, approximating copy levels more than 2× greater than that of wild-type; gain, approximating 1–2 additional copies; SNV, activating promoter mutations in the case of *TERT* and inactivating mutation by indel/nonsense/nonstop in the case of *NF1*; SV breakpoint within gene body or within 100 kb of gene start; or none of the above, i.e., "unaligned"). *P* values by *t*-test using log2-transformed data. Box plots represent 5% (lower whisker), 25% (lower box), 50% (median), 75% (upper box), and 95% (upper whisker). **b** For tumors with SSV breakpoint within 100 kb of *TERT* or *MYB*, respectively, allele-specific patterns associated with increased expression. *P* values by binomial test using cis-X[52]. **c** As compared to all SSVs, fractions of SSVs involving topologically associated domain (TAD) disruption and altered gene expression (defined as FDR < 10% for the gene by distance metric method[4] using 1 Mb region window, with corrections for tumor type and CNA, and expression > 0.4 SD or < −4 SD from median for the case harboring the breakpoint). *P* values by chi-squared test. **d** Percentages of SSV breakpoint associations involving the translocation of an enhancer within 0.5 Mb of the SSV breakpoint in proximity to the gene (and closer than any enhancer within 1 Mb of the unaltered gene), as tabulated for the entire set of SSV breakpoint associations with breakpoint mate on the distal side from the gene, as well as for the subsets of SSV breakpoint associations involving altered gene expression (defined as for **b**). *P* values by chi-squared test. **e** By gene and by histologic type, the number of SV breakpoint associations involving the translocation of an enhancer, which involved 103 genes and 140 tumors. **f** Depiction of the *TERT* locus (left) and *MYB* locus (right) and associated TADs and SSVs. Top: TADs as Hi-C-based contact maps, with gray shading indicating locus interactions (darker, stronger interactions). Bottom: gene expression levels of *TERT* and *MYB* corresponding to SSV breakpoints located in the genomic region. SSV breakpoints are annotated as TAD-preserving (i.e., both breakpoints fall within the same TAD) or TAD-disrupting. Dotted lines denote breakpoints within the same sample and solid lines denote common SV event.

Data 2). More SSV-associated overexpression events involved TAD disruption than enhancer hijacking, with, for example, just one tumor showing *TERT*-associated enhancer hijacking (Fig. 3e) but nine tumors with TERT-associated TAD disruption (Fig. 3f). Significant SSV-gene associations involved all SSV classes and sizes (Supplementary Fig. 2d, e), various tumor statuses (initial, recurrent, progressive, or second malignancy, Supplementary Fig. 2f), and all tumor types (Supplementary Fig. 2g).

Using molecular subtype instead of histologic type in the SSV-expression modeling, or including only tumors originating from brain-specific cell types in the analysis, yielded very similar SSV-gene associations uncovered above using the full cohort of 854 tumors (Supplementary Fig. 3). Based on unsupervised clustering of RNA-seq data, tumors broadly segregated according to histologic type (Supplementary Fig. 3a). Our statistical models included histologic type of the tumor as one of the covariates when defining genes altered in association with nearby SSV breakpoints. Still, the histologic type covariate did not heavily influence the top results (Supplementary Fig. 3b). While many histologic types may be comprised of molecular subtypes, in a global molecular analysis, the most-represented histologic types—including PLGG, PHGG, MBL, EPMT, and ATRT—all formed fairly homogenous groups (Supplementary Fig. 3a). Consistent with the approach of other studies of SSVs and expression[3,10,27], our pan-cancer analyses used histologic classification rather than molecular subtype. Furthermore, the CBTTC's regulatory for CNS tumors intentionally allowed for the broad collection of abnormal cell growth, not necessarily specific to brain-specific cell types. Likewise, all data from 854 tumors that were available from CBTTC formed the basis of our study. However, in an alternate analysis, we first removed tumor types that originated from cell types not specific to the brain, along with metastatic secondary tumors and tumors that did not fit within the more common CBTTC tumor type designations. The remaining set of 793 tumors yielded very similar SSV-expression associations to those of the 854-tumor set (Supplementary Fig. 3c and Supplementary Data 2).

**SSVs involved with gene fusions or pathway alterations.** SSV breakpoints falling within genes and associated with their increased expression may represent gene fusions. We found a highly significant degree of overlapping gene-to-tumor associations involving predicted fusions using RNA-seq chimeric reads, gene overexpression, and SSVs breakpoint falling within the boundary of a gene (Supplementary Fig. 4a). This concordance between orthogonal results sets suggested a path by which we could refine RNA-seq-based fusion predictions using SSV data. Out of 19,211 candidate fusion events identified by RNA-seq analysis (using Arriba or STAR-fusion algorithms), 1866 corresponded to SSV breakpoints found within one or both genes (Fig. 4a), and 1208 of these involved a high expression association (see Methods). This set of 1208 fusion calls with the highest level of support involved 974 distinct gene fusions, 368 tumors, and 331 patients (Supplementary Data 4), as well as the majority of gene body-associated SSV breakpoints with overexpression (Fig. 4b). Of the 368 tumors, 182 had fusions both detectable in two or more tumors and including a known cancer-associated gene by COSMIC[20] (Fig. 4c). The most prevalent fusion identified was *KIAA1549-BRAF*, in 117 tumors, most of them PLGG[28,29], followed by *C11orf95-RELA*, in 22 tumors, predominantly involving EPMT[30]. Other fusions included *CLIP1-ROS1* (n = 4 tumors), *TRIM24-BRAF* (n = 3)[31], *FGFR1-TACC1* (n = 3)[32], *FGFR2-SHTN1* (n = 2)[33], and *MYB-QKI* (n = 2)[34]. RNA-seq-based fusion predictions with SSV support were highly enriched for in-frame fusions (Fig. 4c and Supplementary Fig. 4b). Furthermore, when considering 74 different fusions across 66 tumors from 27 patients for which CBTTC profiled multiple tumors, 52 of these fusions appeared ubiquitous[35] for a given patient (Supplementary Fig. 4c), suggesting that these events might occur earlier in the disease process.

We found SSVs to considerably extend the numbers of pediatric brain tumors somatically altered for critical pathways. Taking a set of cancer-associated pathways and related genes previously annotated based on domain knowledge[19,21,36,37], we examined the CBTTC tumors for alteration in these pathways. Alterations considered were gene fusion, SSV-mediated altered cis-regulation or gene disruption (taking from the genes significant for 1 Mb region), SNV or indel, and deep deletion or high-level amplification (Supplementary Data 5). Across many different brain tumor types, SSV-mediated alterations (Fig. 5a) involved RTK pathway-related genes (*KRAS*, *MET*, *EGFR*, *NF1*), p53/Rb-related genes (*CDK4*), *TERT*, MYC family genes (*MYC*, *MYCN*, *MYB*), SWI/SNF complex genes (*SMARCB1*), and HIPPO pathway-related genes (*NF2*). Across the entire CBTTC cohort, assessment of genes within pathways (Fig. 5b) demonstrated a high number of alterations involving Receptor Tyrosine Kinases (RTKs, 39.1% of tumors), p53 or Rb (14.6%), PI3K/AKT/mTOR (12.3%), chromatin modification (8.8%), TERT (8.1%), MYC family (8.0%), SWI/SNF complex (8.1%), Wnt/beta-catenin (5.7%), and HIPPO signaling (2.9%). We found the above pathways altered in different ways involving different genes in different tumor types (Fig. 5c and Supplementary Fig. 5). Some

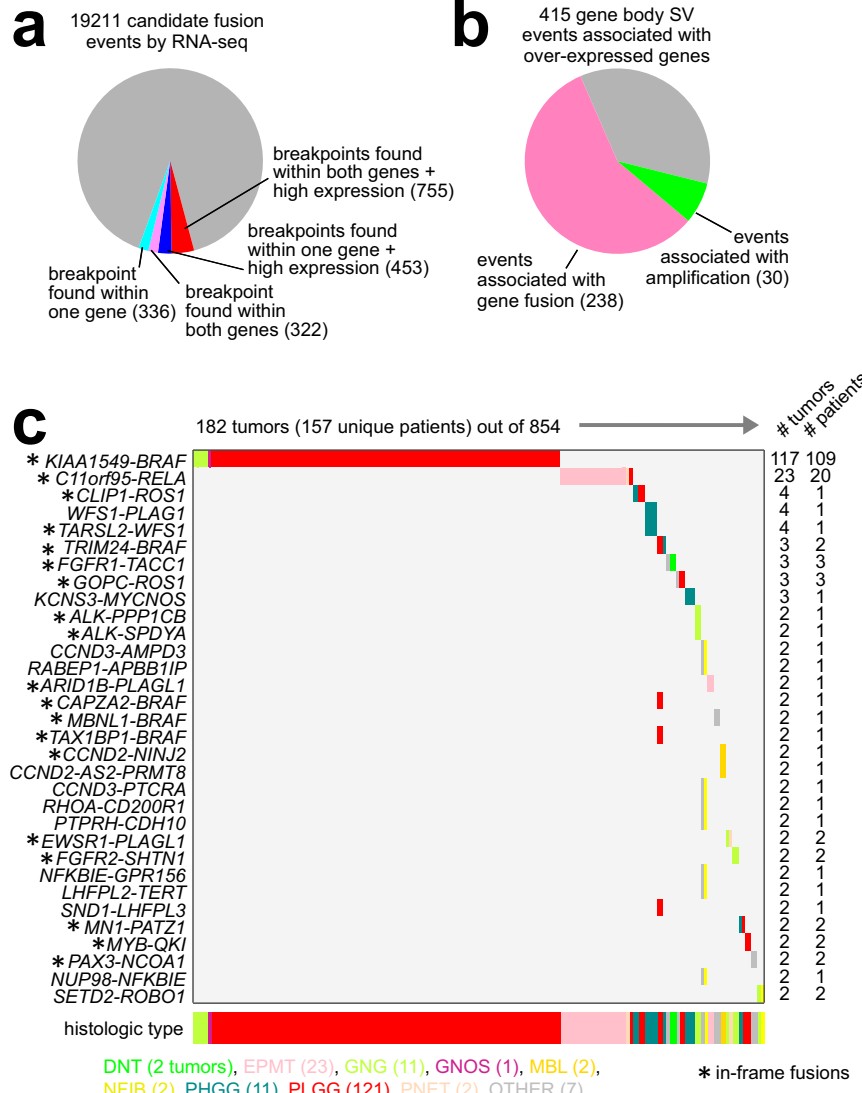

**Fig. 4 Identification of gene fusion events by both RNA-seq and SSVs. a** Out of 19,211 candidate fusion events identified by RNA-seq analysis (using Arriba or STAR-fusion algorithms), numbers of events with support from SSV analyses. As indicated, for 1866 candidate fusion events, SSV breakpoints were found within one or both genes, with or without a high expression association (see Methods). **b** Of the 415 gene body SSV breakpoint events associated with overexpressed genes (70 genes with FDR < 10% correcting for tumor type and CNA), the fractions of events associated with either gene fusion by combined RNA-seq and SSV analysis or high-level gene amplification are indicated. **c** Gene fusions with both RNA-seq and SSV support with high expression association, and involving either greater than two tumors (*KCNS3-MYCNOS*, *TARSL2-WFS1*) or greater than one tumor plus a COSMIC gene[20] are represented. Tumor type is indicated along the bottom and in the coloring of the fusion event.

tumor types showed particularly high enrichment for alterations affecting a specific pathway (Fig. 5b). RTK-related alterations showed enrichment within PLGG[17]; p53/Rb-related alterations, within PHGG[17]; alterations involving chromatin modifiers, TERT, and MYC family, within MBL[10]; SWI/SNF alterations, within ATRT[14]; Wnt/beta-catenin alterations, within CRANIO[38]; and HIPPO pathway alterations, within MNG[39]. SSVs in particular impacted a substantial number of tumors in relation to RTK (186 out of 334 total impacted tumors), p53-related (30 out of 125), TERT (45 out of 69), MYC family (35 out of 68), HIPPO (9 out of 25), and SWI/SNF complex (4 out of 69).

**SSV associations with progressive or recurrent disease.** Of the 854 CBTTC tumor profiles, 174 were from progressive or recurrent tumors and 633 were from initial tumors. These data provided an opportunity to study SSV-mediated alterations specifically occurring within more advanced disease. In a paired

analysis involving 44 patients, increased numbers of SSVs were detected in recurrent or progressive tumors from a given patient, as compared to the initial tumor from the same patient (Fig. 6a, $p = 0.007$, paired $t$-test), suggesting that molecular differences between the two groups would involve SSVs. In addition to identifying SSV-gene associations across the entire set of tumors as described above, we applied our analytical approach separately to the full subset of initial tumors ($n = 633$) and to the full subset ($n = 174$) of recurrent or progressive tumors (using 1 Mb region window, Supplementary Data 6). Both progressive/recurrent and initial tumor subgroups yielded hundreds of genes—318 genes and 249 genes, respectively—with differential expression associated with nearby SSV breakpoints (FDR < 10%[23], correcting for tumor type and CNA, Fig. 6b).

Of the 318 genes significant (FDR < 10%) for the progressive/recurrent tumor group, 222 were not significant for either the initial tumor group or the entire tumor set (Fig. 6b, c), these genes

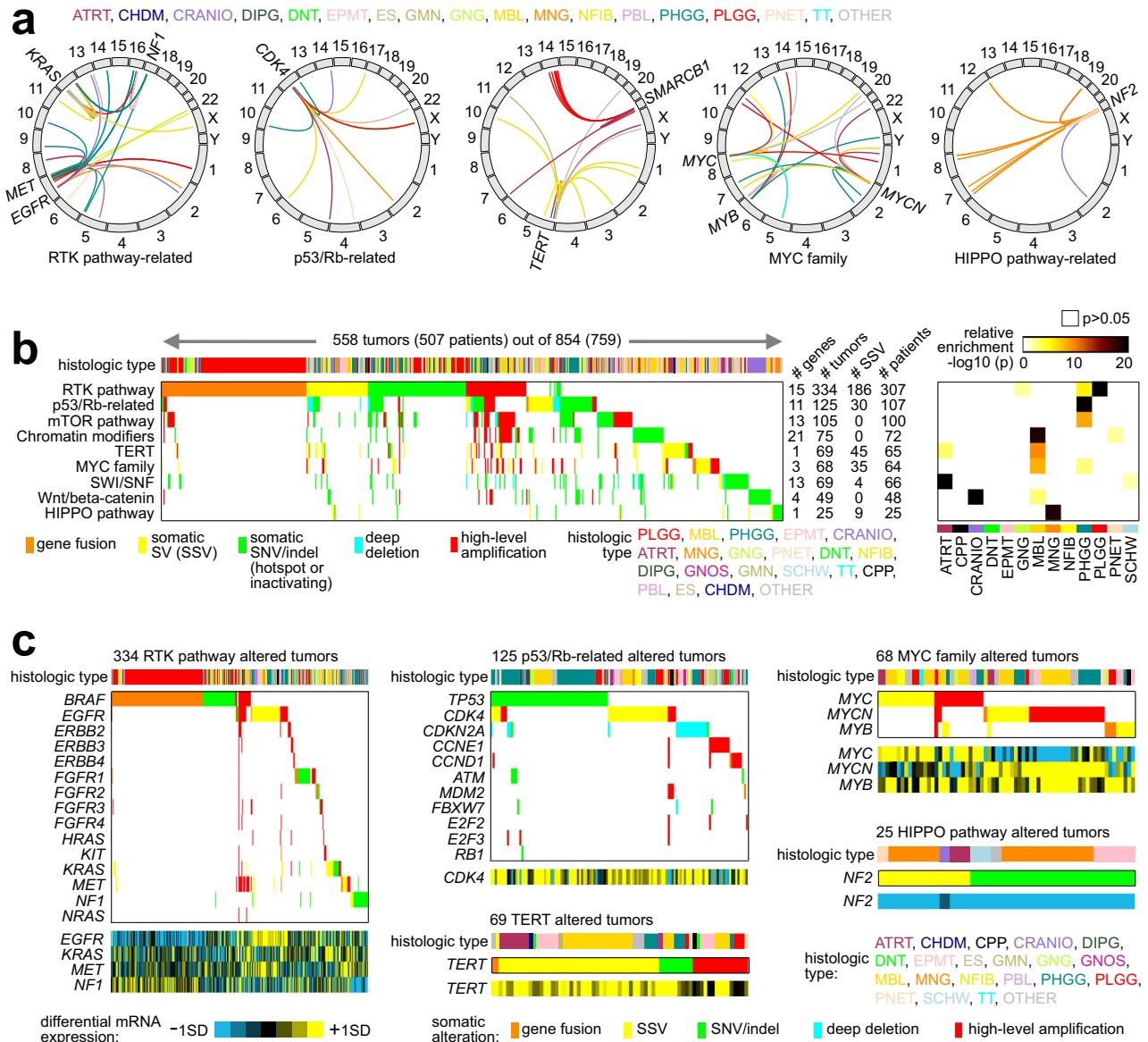

**Fig. 5 SSVs associated with key oncogenic or tumor-suppressive pathways. a** Genomic rearrangements (represented in circos plots) involving Receptor Tyrosine Kinase (RTK) pathway-related genes (*KRAS, MET, EGFR, NF1*), p53/Rb-related genes (*CDK4*), *TERT*, SWI/SNF (*SMARCB1*), MYC family genes (*MYC, MYCN, MYB*), and HIPPO pathway-related genes (*NF2*). SSV events are colored according to tumor type, as indicated. **b** Pathway-centric view of somatic alterations in pediatric brain tumors (representing 558 CBTTC tumors and 507 patients with at least one somatic alteration in the indicated pathways), involving key pathways and genes previously annotated across multiple cancer types based on domain knowledge[19,21,36,37]. Panel on the right represents the significance of enrichment (one-sided Fisher's exact test) of gene alteration events for each pathway within any particular tumor type versus the rest of the tumors (focusing on the 13 tumor types with the most tumors). **c** For the pathways from **b** that also involve at least one SSV event, somatic alteration events involving each gene included in the pathway are represented. For SSV-impacted genes the corresponding differential mRNA expression patterns are shown. For **a**, **b**, events are colored according to the type of somatic alteration: gene fusion, SSV (for oncogenes, breakpoint falling with 1 Mb of gene and associated with expression >0.4 SD from median for the given tumor; for tumor suppressors, breakpoint falling within the gene body and expression < −0.4 SD), SNV or indel (for oncogenes, SNV within hotspot residue[49]; for tumor suppressor genes, SNV within hotspot residue or inactivating mutation by indel/nonsense/nonstop), and deep deletion or high-level amplification (respectively approximating total copy loss and copy levels more than 2× greater than that of wild-type, based on thresholded values).

including *CCND2* (cyclin D2) and *NTRK1*. In addition to issues involving sample power and sensitivity, the differences in the associations made between tumor subsets could involve SSV events being randomly distributed among initial tumor and primary/recurrent groups. To enrich for SSV-gene associations that would be truly specific to progressive or recurrent disease, we first compared the frequency of SSV breakpoint events for each gene (using 100 kb region window) of progressive or recurrent tumors with that of initial tumors. We then overlapped the set of

genes with breakpoints being enriched within progressive or recurrent tumors ($p < 0.05$, one-sided Fisher's exact test) with the set of genes having SSV-gene associations for the same tumor group. The overlap of 34 genes between the two gene sets was statistically significant ($p = 0.003$, one-sided Fisher's exact test, Fig. 6d). The 34 genes included seven COSMIC[20] genes (*TERT, CCND2, PLAG1, ROS1, WT1, KRAS, NTHL1*), which also represented a significant overlap ($p < 0.0005$, one-sided Fisher's exact test). Other genes, such as *NGF* and *FGF6*, encode growth

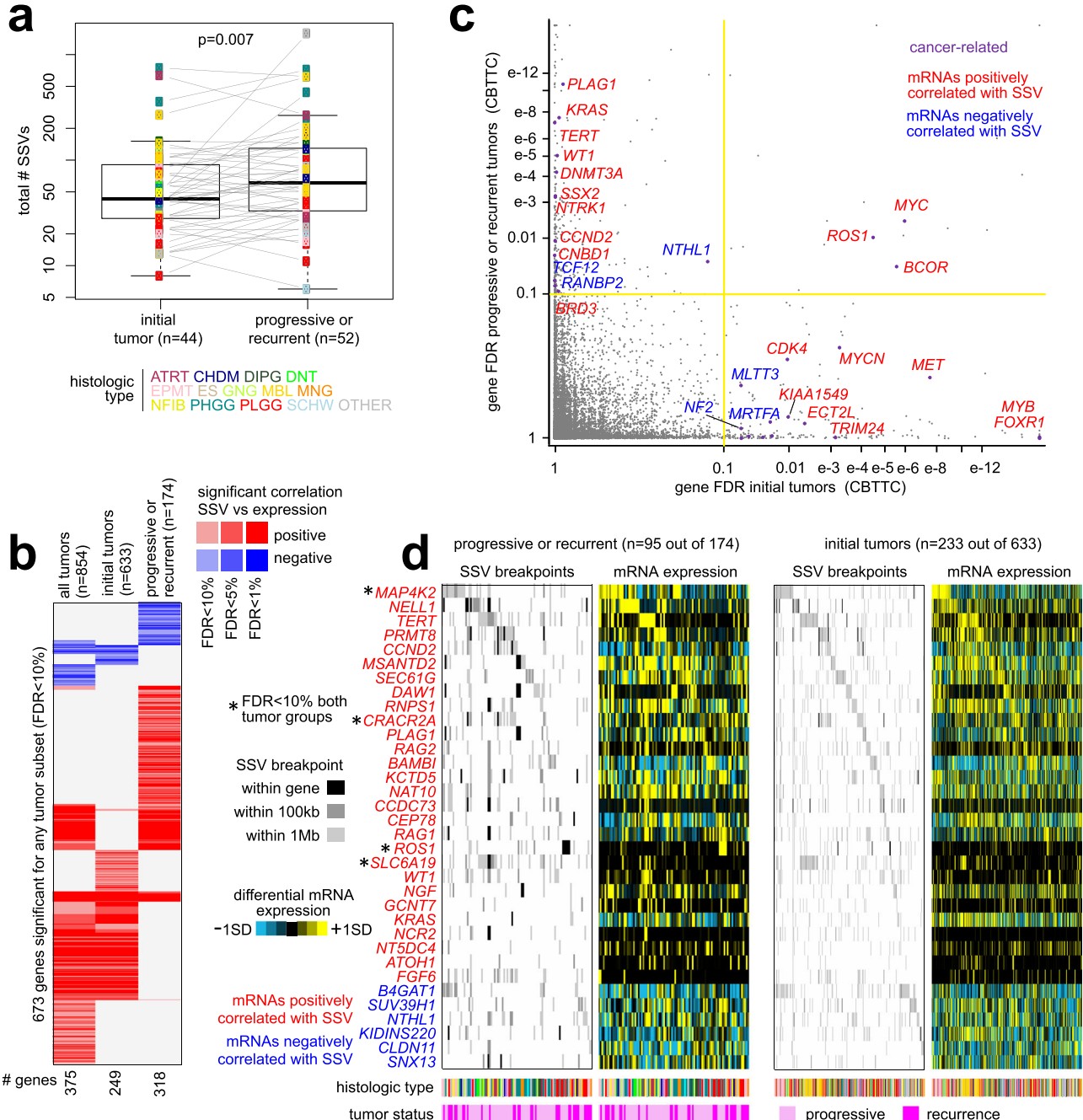

**Fig. 6 SSV-impacted genes associated with progressive or recurrent disease. a** Significant differences between initial tumor and recurrent or progressive tumors for total number of SSVs detected, based on a paired analysis involving 44 patients for which both an initial tumor and a recurrent or progressive tumor were profiled by WGS. *P* values by paired *t*-test on log-transformed data. Box plot represents 5% (lower whisker), 25% (lower box), 50% (median), 75% (upper box), and 95% (upper whisker). **b** Heat map of significance patterns for 673 genes significantly associated with nearby SSV breakpoints (FDR < 10%) for any one of three groups of tumors from CBTTC cohort: (1) all tumors (*n* = 854), (2) initial tumors (*n* = 633), and (3) recurrent or progressive tumors (*n* = 174). Significance by 1 Mb region window and distance metric method[4] (correcting for both tumor type and CNA). Red denotes significant positive correlation; blue, significant negative correlation. **c** Significance of genes in the subset of initial tumors (*n* = 633 tumors, *x*-axis), as compared to their significance in the subset of progressive or recurrent tumors (174 tumors, *y*-axis). Genes in the upper left quadrant reached significance only within the progressive or recurrent tumors. **d** SSV breakpoint and mRNA expression patterns for a set of 34 genes with both a significant SSV-expression association in the subset of progressive or recurrent tumors (**b**) and with significant enrichment within progressive or recurrent tumors (*p* < 0.05, one-sided Fisher's exact test), as compared to the initial tumors, of SSV breakpoints within 100 kb of the gene. For each of the two tumor subsets represented, the ordering of tumors is the same between the SSV breakpoint and expression matrices. Genes with significant SSV-gene association for both progressive/recurrent and initial tumor groups are indicated with an asterisk. See (**a**) for tumor type coloring scheme.

factors. Cumulatively, the 34 genes and associated SSV break-points (using 1 Mb window) and gene expression changes involved 95 out of the 174 progressive or recurrent tumors (55%). These genes also involved 233 out of 633 initial tumors (37%), though representing a smaller proportion of those tumors.

**Molecular alterations involving the overall burden of structural variation.** As another line of investigation, we examined gene expression and DNA mutation features that correlated with the total number of SSV breakpoints detected per tumor, independently of where the breakpoints fall in proximity to genes. Increased numbers of SSVs detected in a tumor from a patient associated with worse patient survival (Fig. 7a, Cox $p = 0.002$, correcting for tumor type). This observation was consistent with previous observations in adult tumors involving SV or CNA burden[4,40] and supports the notion that tumors with high SSV burden have biological differences from other tumors. We searched for genes with inactivating SNVs or indels correlated with high SSV burden, with 17 genes being significant (Fig. 7b, FDR < 5%, Pearson's correlation). While we might not expect all of the 17 genes to have a causative role in SSV burden, two genes of interest included *TP53* and *H3F3C*, as explored below. As a well-known guardian of genome integrity[41], a role of p53 in preventing the type of DNA damage represented by double-strand breaks and SSVs would seem clear. *TP53* hotspot SNVs and inactivating SNVs and indels, along with single-copy loss, were associated with higher SSV burden, in both the CBTTC pediatric brain and TCGA adult pan-cancer cohorts (Fig. 7c). Although not perfectly correlated with SSV burden, the overall burden of SNVs and indels represents another measure of extent of DNA damage (Supplementary Fig. 6a). *TP53* mutation also corresponded to increases in detected SNVs and indels, in addition to SSV burden (Supplementary Figs. 6b), with mutation and copy loss events mostly occurring in PHGG and MBL tumors.

We found thousands of gene expression correlates of overall SSV burden levels (Fig. 7d and Supplementary Data 7), with 2381 genes either increased or decreased (FDR < 5% by linear regression model, correcting for tumor type and gene-level CNA). We observed highly significant overlapping gene patterns between the respective expression signatures of SSV burden between pediatric brain cancers and adult pan-cancers (Supplementary Fig. 6c), indicative of common processes at work across multiple tumor types. We compared enriched gene categories among the respective SSV burden association signatures from pediatric brain tumor, adult glioma (Supplementary Fig. 6d), and adult pan-cancer. All three signatures showed high enrichment for genes related to cell cycle process, chromosome organization, cell division, DNA repair including double-strand break repair, and telomere organization (Fig. 7e). The associations involving DNA double-strand break repair pathway, in particular, were also evident when examining essential individual genes, including *BRCA1*, *BRCA2*, *CHEK2*, *FANCB*, *FANCD2*, *FANCI*, and *RAD51*, and *XRCC2* (Fig. 7f and Supplementary Fig. 6e). On the other hand, the p53/Rb1 pathways appeared altered differently among the different tumor types at the gene expression level, with pediatric brain tumors showing high *MDM2* expression with high SSV burden and with adult pan-cancers instead showing high expression of *CCNE1* and *E2F3*, though with both sets of tumors showing high *CDK4* (Fig. 7f). Notably, however, a previously noted decrease in gene signatures of immune cell infiltrates and in immune response-related genes[4] was observed here only for adult non-brain tumors (Fig. 7f).

A survey of Histone H3 genes showed frequent mutations in *H3F3A* and *H3F3C* across CBTTC tumors (Supplementary Fig. 7a). While *H3F3A* mutations are commonly associated with

pediatric brain tumors, particularly with PHGG and DIPG[28,42], *H3F3C*, which also encodes for an H3.3 histone component, appears to be much less studied in the context of cancer, including pediatric brain cancer. *H3F3C* was uniquely associated with SSV burden over thousands of other genes tested (Fig. 7b), with no hypermutated tumors having *H3F3C* mutations. While *H3F3A* mutations in CBTTC tumors primarily involved the known K28M/K27M[43] hotspot ($n = 30$ tumors), followed by the G35R/G34R[43] hotspot ($n = 4$), *H3F3C* mutations followed another distinctive pattern. All nine impacted tumors (three PLGG, two EPMT, and one each for ATRT, MNG, PHGG, and SARCNOS; two progressive, two recurrent, and one second malignancy) had nucleotide changes involving both K37 duplication and N79K amino acid change (Fig. 8a). Two of the nine tumors also harbored amino acid changes L104F and V89I. *H3F3C* mutations were mutually exclusive with TP53 alterations (Fig. 8b). Relative to other cancers, *H3F3C*-mutated CBTTC tumors showed a very high number of detected SSVs (Fig. 8b, c) as well as SNVs and indels (Supplementary Fig. 7b), reflecting the role of Histone H3.3 in maintaining genome integrity[44]. *H3F3C*-mutated CBTTC tumors also showed lower *H3F3C* mRNA expression versus wild-type ($p < 0.03$, $t$-test on log-transformed data). We went on to survey whole-exome sequencing data from 10,224 TCGA adult cancers[21] for *H3F3C* mutations. Of all TCGA tumors, 49—representing many different tissues of origin—harbored a mutation in *H3F3C* (Fig. 8d), and these tumors also showed a high mutation rate (Fig. 8e). *H3F3C*-mutated tumors ($n = 7$) in PCAWG WGS dataset, also showed increased SSV burden (Supplementary Fig. 7c). The TCGA *H3F3C* mutations did not show the same tight hotspot pattern found in CBTTC tumors, although TCGA mutations did include G35R (two patients), N79K (one patient), and V89I (four patients).

## Discussion

Using the CBTTC pediatric brain tumor datasets of WGS and gene expression, we have shown here how genomic rearrangements, leading to altered gene regulation or gene disruption, globally impact large numbers of genes and patients. We found that the overall phenomenon of SSV-mediated cis-regulatory alterations, as previously observed in adult cancers of various types represented in the PCAWG and TCGA datasets[3,4,19], is also at work in pediatric brain tumors of various types. Aspects of this phenomenon, as observed in both PCAWG and CBTTC cohorts, include: hundreds of genes recurrently impacted, SSV breakpoints as far as 1 Mb from the gene contributing to deregulation, rearrangements involving widespread CNA patterns, many more genes with increased over decreased expression associated with SSV breakpoints, and overexpressed and under-expressed genes respectively representing known oncogenes and tumor suppressor genes. At the same time, the specific set of genes deregulated by SSVs differed considerably between pediatric brain and adult cancers, including adult gliomas. Our study provides a rich resource, whereby the SSV-mRNA associations uncovered here may be further explored to establish novel pediatric brain cancer drivers and mechanistic links with cancer phenotypes. This relatively new class of genomic alterations involving noncoding regions would also have implications for personalized and precision medicine approaches, which at present may focus more on alterations within the exome.

The CBTTC datasets provided unique opportunities for us to explore SSV-mediated cis-regulatory alterations across pediatric brain tumors of diverse histologic types. Previous genomic studies involving pediatric brain tumors have explored the landscape of SSVs, but without corresponding expression data to link the SSV breakpoints with transcription of the nearby genes. A recent

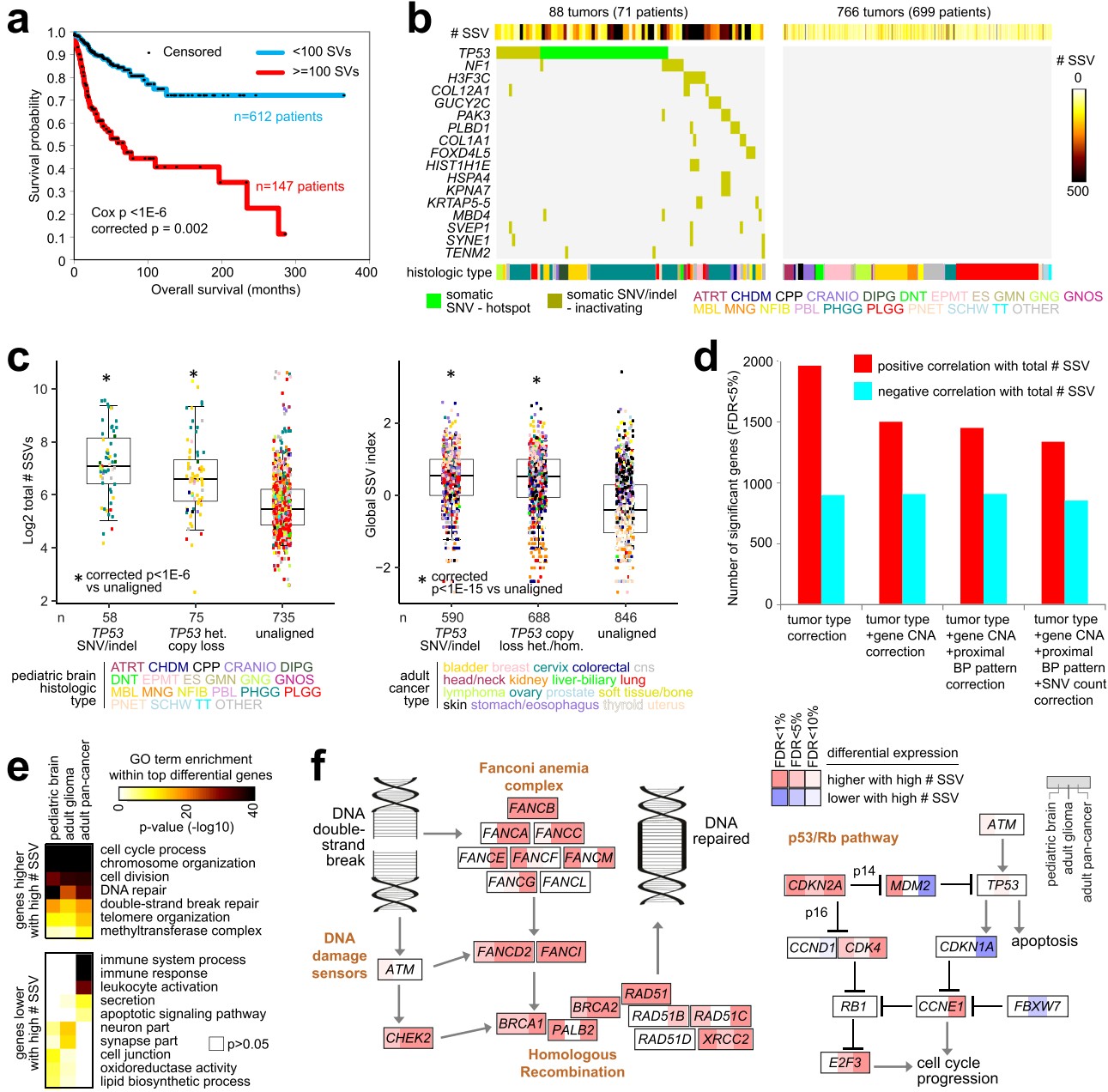

**Fig. 7 Molecular alterations associated with the overall burden of structural variation across pediatric brain tumors. a** Association of the total number of SSVs detected with patient survival ($n = 759$ patients, one tumor per patient). As indicated, $p$ values are by either univariate Cox or stratified Cox according to tumor type ("corrected"), using log-transformed SSVs numbers. **b** Genes for which inactivating SNVs or indels were associated with increasing numbers of SSVs in CBTTC tumors (FDR < 5%, Pearson's correlation, and mutation events in at least three tumors). **c** For both CBTTC cohort (left) and TCGA adult pan-cancer cohort (right), association of TP53 mutation or copy loss with the overall burden of SSVs. $P$ values by linear model correcting for tumor type. Box plots represent 5% (lower whisker), 25% (lower box), 50% (median), 75% (upper box), and 95% (upper whisker). **d** Numbers of significant genes (FDR < 5%), showing a correlation between expression and the total number of SSV events detected across the 854 CBTTC tumors. Linear regression models evaluated significant associations when correcting for specific covariates, as indicated. Proximal BP pattern, SSV breakpoint pattern in relation to the given gene. SNV count based on exome analysis. **e** Selected significantly enriched Gene Ontology (GO) terms for genes correlated (FDR < 10%, with corrections for tumor type and gene-level CNA) with the total number of SSV events, with enrichment patterns (as indicated by degree of shading) evaluated separately for CBTTC pediatric brain, TCGA adult glioma, and TCGA-ICGC adult pan-cancer. $P$ values by one-sided Fisher's exact test. **f** Diagram of key genes involved with DNA double-strand break repair[4] and p53/Rb[36] pathways, with corresponding correlations with the overall structural variation burden within CBTTC pediatric brain, TCGA adult glioma, and TCGA-ICGC adult pan-cancer datasets (red, significantly higher with an increasing number of SSVs, correcting for tumor type and CNA).

study of MBL[10] did feature both expression and WGS data on ~164 tumors. However, our present pan-cancer study could identify SSV-gene associations involving tumors from multiple histologic types. Some SSV-gene associations were significant in the analysis of individual tumor types did not reach significance when analyzing the combined pan-CBTTC set, e.g., associations involving *GFI1* and *PRDM6* as previously observed in MBL. Most significant genes involved cases spanning more than one

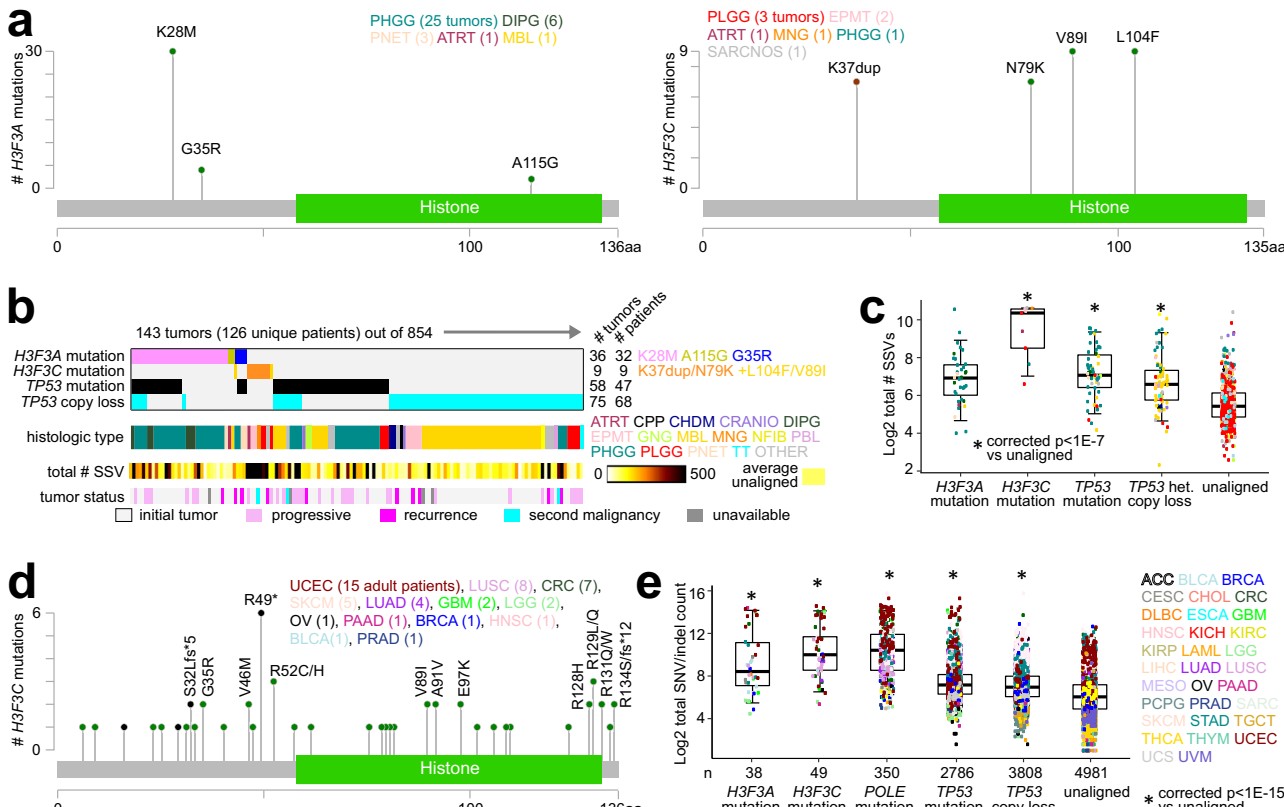

**Fig. 8 Histone H3.3 somatic alterations in pediatric brain tumors include *H3F3C* mutations. a** Prevalence and distribution of somatic SNVs or indels within *H3F3A* (left, involving 36 tumors and 32 patients) and *H3F3C* (right, involving nine tumors and nine patients), based on CBTCC tumors. **b** Across the 143 impacted CBTTC tumors, somatic mutations involving *H3F3A*, *H3F3C*, and *TP53*, as well as *TP53* copy loss. **c** For CBTTC cohort, associations of somatic alterations involving *H3F3A*, *H3F3C*, or *TP53* with overall burden of SSVs. Numbers of tumors in each group from **c**, with 711 tumors in "unaligned" group. *P* values by linear model correcting for tumor type. **d** Prevalence and distribution of somatic SNVs or indels within *H3F3C* in TCGA adult pan-cancer cohort involving 49 patients (out of 10224 with whole-exome sequencing). **e** For TCGA cohort, associations of somatic alterations involving *H3F3A*, *H3F3C*, *POLE*[56], or *TP53* with the total number of detected exome SNV/indel mutations. *P* values by linear model correcting for tumor type. See Methods regarding tumor type abbreviations. Box plots represent 5% (lower whisker), 25% (lower box), 50% (median), 75% (upper box), and 95% (upper whisker).

histologic type, and a few key genes, including *EGFR* and *MYCN* were significant only in the pan-cancer analysis and not associated with any individual tumor type, which highlights the utility of our pan-cancer approach. While several associations found in the present study would be consistent with previous studies' findings, we find hundreds of other genes being impacted by SSVs using our systematic analytical approaches.

Our study identified SSV associations specifically within progressive or recurrent disease, as compared to the initial tumors. This result involves another unique aspect of the CBTTC datasets, which includes patients with multiple tumors profiled. In our present study, we found that different tumors from the same patient tended to demonstrate extensive molecular heterogeneity among them, to the extent that each tumor could be regarded as a separate disease entity in the molecular analyses. Our observations would be consistent with those of previous studies. For example, in MBL studies, genetic events in recurrent tumors exhibited a very poor overlap (<5%) with those in the matched initial tumors[45]. In contrast, most other cancer genomics studies to date have taken a single tumor profile to be representative of the patient's disease. For a given patient, the overall numbers of SSVs and other somatic mutations tend to increase in a recurrent or progressive tumor as compared to the primary tumor. However, finding SSV-mediated expression changes between initial and recurrent tumor in a paired analysis may be challenging with

the current dataset, as the numbers of tumors involved are relatively few, while the SSV events involving any given gene tend to be sparse across the entire CBTTC cohort. The CBTTC cohort may not necessarily lend itself to studies of intratumoral heterogeneity[35], as in CBTTC multiple initial tumors from the same patient are profiled, rather than multiple samples taken from the same initial tumor.

Our study also identified molecular correlates of high overall SSV burden in pediatric brain cancer. Analogous to results from adult cancers, transcriptional programs associated with pediatric brain tumors having high numbers of genomic rearrangements involved DNA damage response and cell proliferation. DNA mutation correlates of high SSV burden included mutations in *TP53* and histone H3.3 genes. Our *TP53*-related findings would be consistent with those of an adult pan-cancer study of TCGA data[46], in which *TP53* mutational status was associated with global increases in DNA copy number instability and somatic SNV/indel frequency. Other studies have also linked *TP53* mutation with increased numbers of chromosome rearrangements in pediatric cancers[15,47]. While *H3F3A* mutations are commonly associated with pediatric brain cancers, *H3F3C* appears to be much less studied in cancer, including pediatric brain cancer. Interestingly, a search of the PeCan (https://pecan.stjude.cloud/) and PedCBioportal (https://pedcbioportal.kidsfirstdrc.org/) databases—representing more than 1000

additional pediatric brain tumors—did not uncover additional cases of *H3F3C* hotspot mutation. This may be because mutations in CBTTC cohort were few and spread among multiple histologic types and involving progressive or recurrent or second malignancy cases. At the same time, the observations in adult tumors would suggest that alteration of histone H3.3 genes (including *H3F3C*) may cumulatively involve many patients, with the functional impact not being limited to hotspot mutations. Our study demonstrated that combined genomic analyses utilizing CBTTC and PCAWG data could identify patterns occurring across adult and pediatric cancers and patterns unique to pediatric brain cancer.

## Methods

**Patient cohorts.** Results are based upon data generated by the CBTTC. Combined WGS analysis (at ×60 coverage) and RNA-seq analysis (at ×30 coverage) was carried out for 854 pediatric brain tumor samples in total, representing 759 patients. Tumor samples in CBTTC spanned at least 33 different tumor types: adenoma; ATRT; chordoma; neurocytoma; choroid plexus carcinoma; CPP; CRANIO; DIPG; DNT; EPM, subependymal giant cell astrocytoma; EPMT; ES; GMN; ganglioneuroblastoma; GNG; GNOS, glial-neuronal tumor not otherwise specified (NOS); HMBL; LCH; MBL; MNG; MPNST; NBL; NFIB; oligodendroglioma; pineoblastoma; primary CNS lymphoma; PHGG (WHO grade III/IV); PLGG (WHO grade I/II); PNET, supratentorial or spinal cord primitive neuroectodermal; rhabdomyosarcoma; SARCNOS; SCHW; teratoma; and other/unspecified. The histologic designations of the tumors, as provided by the individual CBTTC member institutions contributing the samples, were confirmed by independent pathology review at the CBTTC centralized biorepository, with the majority of contributing sites providing representative histology slides. Tumor molecular profiling data were generated through informed consent as part of CBTTC efforts and analyzed here per CBTTC's data use guidelines and restrictions.

A subset of CBTTC tumors represents multiple tumors taken from the same patient, involving 170 tumor samples from 75 patients in total. As indicated in Supplementary Data 1, multiple tumors from the same patient may entail samples from multiple initial tumors, or samples taken at different times, e.g., samples taken initially from the initial tumor and later from a progressive or recurrent tumor. Different tumors from the same patient often demonstrated extensive molecular heterogeneity with respect to each other (Fig. 1c and Supplementary Fig. 1c). Therefore, each tumor sample was analyzed independently in the integrative analyses, with the numbers of patients and tumors involved with a particular pattern of interest noted where warranted.

The results here are also based in part upon data generated by both TCGA Research Network and the ICGC. Previously, we carried out combined WGS and RNA-seq analysis for 2334 TCGA-ICGC cancer cases in total[4], 1892 of which were from TCGA and 1232 of which (including all ICGC cases and 790 TCGA cases) were part of the PCAWG consortium efforts. Of the 2334 cases, 25 involved patients under the age of 18, of which 21 were ICGC lymphomas. Cases profiled spanned a range of tumor types (bladder, sarcoma, breast, liver-biliary, cervix, leukemia, colorectal, lymphoma, prostate, eosophagus, stomach, central nervous system or "cns", head/neck, kidney, lung, skin, ovary, pancreas, thyroid, uterus). We aggregated molecular data from public repositories. Tumors in TCGA spanned 32 different TCGA projects, each project representing a specific tumor type, listed as follows: LAML, acute myeloid leukemia; ACC, adrenocortical carcinoma; BLCA, bladder urothelial carcinoma; LGG, lower grade glioma; BRCA, breast invasive carcinoma; CESC, cervical squamous cell carcinoma and endocervical adenocarcinoma; CHOL, cholangiocarcinoma; CRC, colorectal adenocarcinoma (combining COAD and READ projects); ESCA, esophageal carcinoma; GBM, glioblastoma multiforme; HNSC, head and neck squamous cell carcinoma; KICH, kidney chromophobe; KIRC, kidney renal clear cell carcinoma; KIRP, kidney renal papillary cell carcinoma; LIHC, liver hepatocellular carcinoma; LUAD, lung adenocarcinoma; LUSC, lung squamous cell carcinoma; DLBC, lymphoid neoplasm diffuse large B-cell lymphoma; MESO, mesothelioma; OV, ovarian serous cystadenocarcinoma; PAAD, pancreatic adenocarcinoma; PCPG, pheochromocytoma and paraganglioma; PRAD, prostate adenocarcinoma; SARC, sarcoma; SKCM, skin cutaneous melanoma; STAD, stomach adenocarcinoma; TGCT, testicular germ cell tumors; THYM, thymoma; THCA, thyroid carcinoma; UCS, uterine carcinosarcoma; UCEC, uterine corpus endometrial carcinoma; UVM, uveal melanoma. Tumor molecular profiling data were generated through informed consent as part of previously published studies and analyzed per each original study's data use guidelines and restrictions.

**Molecular profiling datasets.** The somatic DNA workflow for DNA variant calling is available in the KidsFirst Github repository (https://github.com/kids-first/kf-somatic-workflow). CBTTC used Manta SV v1.4.0 algorithm for SSV calls[48] based on WGS data. The hg38 reference for SSV calling used was limited to canonical chromosome regions. We accessed the SSV VCF files through the public project on the Kids First Data Resource Portal and Cavatica (https://cbttc.org/). We used only SSV calls that passed quality filters in the analyses. Manta algorithm

classified each SSV call as one of the following: tandem duplications, insertions, deletions, inversions, and translocations. TCGA-ICGC SSV calls were previously compiled[4] from both PCAWG consortium and internal calling using Meerkat algorithm[6].

CBTTC used both Strelka2 v2.9.3 and Mutect2 v4.1.10 to call simple variants, i.e., SNV and insertions/deletions (INDEL), based on WGS data. We assessed the somatic variant MAFs through the public project on the Kids First Data Resource Portal and Cavatica (https://cbttc.org/). We used only variant calls that passed quality filters in the analyses. Variant calls made by either Strelka2 or Mutect2 were considered, as the focus of this study was on SNVs and indels involving already known cancer genes[21] and hotspot residues[49], and with allowances made for the lower sequencing coverage of WGS as compared to that of whole-exome sequencing (WXS). For TCGA WXS data, we obtained somatic mutation calls from the publicly available "MC3" TCGA MAF file (covering $n = 10224$ patients), available at https://www.synapse.org/#!Synapse:syn7214402; we used variants called by two or more algorithms in this study.

Gene-level CNA calls, made based on CBTTC WGS data, were obtained from the PedCBioPortal (https://pedcbioportal.org/datasets). Low-level gene gain (approximating 3–4 copies), high-level gene amplification (approximating five or more copies), low-level copy loss (approximating heterozygous loss), or deep copy loss (approximating gene deletion) were inferred using the "thresholded" calls (with values of $+1$, $+2$, $−1$, or $−2$, respectively) as made available by PedCBioPortal. For TCGA data, we obtained thresholded gene-level CNA calls, made based on Affymetrix SNP 6.0 arrays, from the Broad Institute Firehose pipeline (http://gdac.broadinstitute.org/).

We obtained processed RNA-seq data for CBTTC tumors from the PedCBioPortal (https://pedcbioportal.org/). RNA-seq data were quantile normalized prior to the analyses. For TCGA glioma cases (GBM and LGG projects), we obtained RNA-seq data from The Broad Institute Firehose pipeline (http://gdac.broadinstitute.org/). All RNA-seq sample profiles were aligned using the by UNC RNA-seq V2 pipeline[50]. We previously carried out integrative analyses of SSV and RNA-seq for TCGA-ICGC cohort[4], using a combined and harmonized RNA-seq dataset across the two cohorts.

**Integrative analyses between SSVs and gene expression.** We defined genes with altered expression associated with nearby SSV breakpoints by two methods, demonstrated previously using TCGA and TCGA-ICGC[3,4,19] datasets. We used our "genomic region window" method for patterns involving SSV breakpoints falling within a given gene or within 100 kb upstream or 100 kb downstream of the gene, and our "distance metric" method involved weighting the relative distances from the gene start within a region ±1 MB surrounding the gene. These analyses using the CBTTC datasets included 18,786 unique named genes. By design[4], our analytical approach does not assume the specific mechanism of altered expression (as many diverse scenarios would be plausible). This aspect would include treating SSV breakpoints representing different classes (tandem duplications, insertions, deletions, inversions, and translocations) and insert sizes the same in the integration with gene expression.

The genomic region window method starts with a set of specified genomic region windows of interest in relation to genes. For each region, we constructed an SSV breakpoint matrix by annotating for every tumor the presence or absence (using "1" or "0", respectively) of at least one SSV breakpoint within the given region. For the set of SSVs associated with a given gene within a specified region in proximity to the gene (e.g., 0–100 kb upstream, 0–100 kb downstream, or within the gene body), correlation between expression of the gene and the presence of an SSV breakpoint was assessed using a linear regression model (with log-transformed expression values). These linear regression models considered genes with at least three tumors associated with an SSV within the given region.

The distance metric method is similar to the genomic region windows approach, but with the [gene × tumor] breakpoint pattern matrix constructed differently. We considered the ±1 Mb region window surrounding each gene (spanning from 1 Mb upstream of the gene start to 1 mb downstream of the gene start). For each tumor, we tabulated the relative distances of the SSV breakpoint closest to the start of each gene, assembling a [gene × tumor] relative distance matrix. For a particular gene in a given tumor having no breakpoints within ±1 Mb, we imputed the maximum distance of 1 Mb. We log2-transformed the absolute relative distances. Using this breakpoint pattern matrix, we assessed the correlation between expression of the gene and the presence of an SSV breakpoint, using a linear regression model (with log-transformed expression values). The distance metric method provides a single result for each gene across the samples, representing genes consistently altered across the entire ±1 Mb region examined. The 1 Mb window was used as genome rearrangements may involve the translocation of enhancers, which may impact genes within a distance of ~1 Mb[4]. The closer that SSV breakpoints are to the gene start in relation to altered expression, as compared to tumors without breakpoints, the more significant the association between expression and SSVs. Breakpoints further away from the given are less weight than breakpoints in closer proximity.

In addition to modeling expression as a function of SSV events, the model incorporated tumor type (as encapsulated by one of the ~33 CBTTC tumor types) as a covariate. Therefore, any significant association between genes and SSV breakpoint patterns must rise above any association that would be explainable by

tumor type alone. Similarly, the models incorporated gene-level CNA (using the thresholded values of +1, +2, −1, or −2) as a covariate, given the observed associations of CNA with nearby SSV breakpoints[3,4,19]. In the downstream analyses, we explored the genes for which SSV associations were significant (FDR < 10%) after correcting for both tumor type and CNA. Also, by distance metric method (including CNA as a covariate), we assessed 13 pediatric brain tumor types separately for global SSV-expression associations, these tumor types (ATRT, CPP, CRANIO, DNT, EPMT, GNG, MBL, MNG, NFIB, PHGG, PLGG, PNET, SCHW) being the ones with the most numbers of tumors in the CBTTC cohort (at least 13 tumors).

**Gene fusion analysis**. Both STAR-Fusion v1.5.0 and Arriba v1.1.0 algorithms were used by CBTTC to make candidate fusion calls based on RNA-seq data to identify chimeric sequencing reads. We assessed these candidate fusion calls through the public project on the Kids First Data Resource Portal and Cavatica (https://cbttc.org/). We removed from further consideration fusions likely to represent artifacts based on membership in the "banned" list by FusionCatcher algorithm (https://github.com/ndaniel/fusioncatcher/blob/master/bin/generate_banned.py), as well as events for which both partners represented the same gene. We refined the remaining 12,624 RNA-seq-based fusion event calls using the SSV data as follows. We gave priority to fusion calls for which SSV breakpoints by WGS fell within one or both of the associated genes, and for which there was a high expression association. A high expression association was defined here as one of the following: (1) for fusion events occurring in one or two tumors, whether for each tumor the expression of either gene was >0.4 SD from the median; or (2) whether a significant association between SSV breakpoints and increased expression ($p < 0.01$, linear model incorporating tumor type and CNA) was found for either gene, either by distance metric method or by genomic region window within the gene body.

**Pathway-level somatic alteration categories**. For the pathway-centric view of somatic alterations in pediatric brain tumors (Fig. 5), key pathways and genes considered included: RTK pathway (*BRAF, EGFR, ERBB2, ERBB3, ERBB4, FGFR1, FGFR2, FGFR3, FGFR4, HRAS, KIT, KRAS, MET, NF1, NRAS*), HIPPO pathway (*NF2, SAV1, WWC1*), chromatin modification (*CREBBP, EHMT1, EHMT2, EP300, EZH1, EZH2, KAT2A, KAT2B, KDM1A, KDM1B, KDM4A, KDM4B, KDM5A, KDM5B, KDM5C, KDM6A, KDM6B, KMT2A, KMT2B, KMT2C, KMT2D, KMT2E, NSD1, SETD2, SMYD4, SRCAP*), SWI/SNF complex (*ACTB, ACTL6A, ACTL6B, ARID1A, ARID1B, ARID2, BCL11A, BCL11B, BCL6, BCL6B, BRD7, BRD9, DPF1, DPF2, DPF3, PBRM1, PHF10, SMARCA2, SMARCA4, SMARCB1, SMARCC1, SMARCC2, SMARCD1, SMARCD2, SMARCD3, SMARCE1*), mTOR pathway (*AKT1, AKT2, AKT3, MTOR, PIK3CA, PIK3R1, PTEN, RHEB, STK11, TSC1, TSC2, IDH1, IDH2, VHL*), MYC family (*MYC, MYCN, MYB*), *TERT*, Wnt/beta-catenin (*APC, AXIN1, CTNNB1, FGF19, NCOR1*), and p53/Rb-related (*ATM, CCND1, CCNE1, CDK4, CDKN1A, CDKN2A, E2F2, E2F3, FBXW7, MDM2, RB1, TP53*). Other pathways considered included the NRF2 pathway (*NFE2L2, KEAP1, CUL3, SIRT1, FH*), but as there were few cases involved in this pathway, we did not include it in the final presentation. For known oncogenes (e.g., *AKT1, MTOR, PIK3CA, RHEB, BRAF, EGFR, ERBB2, ERBB3, HRAS, KRAS, NRAS*), if an SNV occurred in "hotspot" residues as reported by Chang et al.[49], the SNV was considered in the analysis. We considered all inactivating SNVs (nonstop/nonsense) and indels in putative tumor suppressor genes (e.g., *TP53*) in the analyses. We also considered *TERT* activating promoter mutations[51]. At both the gene and pathway levels, we tabulated somatic alterations in the following order: SNV or indel, gene fusion, deep deletion (approximating homozygous loss), high-level amplification (approximating five or more copies), and SSV (for oncogenes, breakpoint falling with 1 Mb of gene and associated with expression >0.4 SD from median for the given tumor; for tumor suppressors, breakpoint falling within the gene body and expression < −0.4 SD). SSVs were considered only for those genes significant (FDR < 10%) by SSV-expression analyses for either the gene body region or the 1MB region (incorporating tumor type and CNA).

**Molecular correlates of the overall extent of genomic rearrangement**. We assessed differential gene expression patterns associated with the overall burden of structural variation across pediatric brain tumors, as we did in our previous study involving adult cancers[4]. SSV calls made for each tumor profiled were tabulated (for translocation SSVs where each breakpoint appeared as a separate entry, the two entries counted as a single SSV). For each gene, we assessed the correlation between expression and the total number of SSV events detected across the 854 CBTTC tumors. We used linear regression models with both log-transformed expression values and log-transformed SSV event numbers, correcting for specific covariates where indicated. We carried out a similar analysis using the TCGA GBM-LGG combined dataset of 107 tumors, though here a correction was added for low pass versus high pass WGS (a technical factor impacting SSV detection involving TCGA datasets[19]). As a measure of total SNVs and indels for a given tumor, we used the "mutation count" field from the clinical data files provided by PedCBioPortal (https://pedcbioportal.org/), which is based on analysis of exomes by CBTTC investigators.

We also assessed genes with inactivating SNVs and indel mutations associated with the overall burden of structural variation. For CBTTC datasets, we constructed a [gene × tumor] matrix, involving 567 genes with nonsense/nonstop/indel mutations in at least three tumors. We assessed the correlation between mutation events (presence/absence represented by 1/0, respectively) and log-transformed SSV numbers by Pearson's correlation, with FDR correction by Storey and Tibshirani[23]. FDR estimation by permutation testing (shuffling the SSV profiles with respect to the expression profiles) was also carried out for comparison with the Storey and Tibshirani estimates, with results provided in Supplementary Fig. 2a and Supplementary Data 2. We then assessed the set of genes significant with FDR < 5% in the TCGA datasets consisting of 1892 adult cancers, by linear model correcting for both tumor type and high pass versus low pass WGS.

**Integrative analyses using TAD and enhancer genomic coordinates**. To identify breakpoints associated with TAD disruption, we used recently published TAD data from the IMR90 cell line[25], following the same approach as described previously[4,19], and using the UCSC Genome Browser LiftOver tool to convert TAD coordinates from hg18 to hg38. We defined TAD-disrupting SSVs as those SSVs for which the two breakpoints did not fall within the same TAD. For specific genes, we used cis-X[52] to evaluate allele-specific expression in relation to SSV-impacted tumors.

For each SSV breakpoint association 0–500 kb upstream of a gene (each association involving unique breakpoint and gene pairing, with only the SSV breakpoint closest to the start of each gene being considered for each tumor in the instance of multiple breakpoints being detected), we determined the potential for translocation of an enhancer near the gene that would be represented by the rearrangement (based on the orientation of the SSV breakpoint mate). We utilized the enhancer annotations as provided by Kumar et al.[26]. We tabulated SSV breakpoint-to-gene associations involving enhancer translocation within 0.5 Mb of the SSV breakpoint in proximity to the gene (assuming no other disruptions involving the region), where the unaltered gene either had no enhancer within 1 Mb or had an enhancer further away from the gene than the translocated enhancer. We considered only SSVs with breakpoints on the distal side from the gene in this analysis. In other words, for genes on the negative strand, the upstream sequence of the breakpoint (denoted as positive orientation) should be fused relative to the breakpoint coordinates, and for genes on the positive strand, the downstream sequence of the breakpoint (denoted as negative orientation) should be fused relative to the breakpoint coordinates.

**Statistical analysis**. All $P$ values were two-sided unless otherwise specified. We utilized linear regression models to associate the expression of genes with nearby SSV breakpoints and with structural variation burden, as described above. In all of the linear models performed in this study, appropriate data transformations were used to make the data align better with the model assumptions. Linear modeling was performed using the lm function in R (version 4.0.3), with permutation testing performed using the shuffle function of the mosaic package. One-sided Fisher's exact tests or chi-squared tests determined significance of overlap between two given feature lists. The method of Storey and Tibshirani[23] estimated FDR for significant genes, using the following formula for each gene: [(nominal $p$ value) × (total number of genes tested)/(total number of genes that were significant at the given $p$ value)]. Visualization using heat maps was performed using JavaTreeview[53] and matrix2png (version 1.2.1)[54]. Enrichment of GO annotation terms within sets of differentially expressed genes was evaluated using SigTerms software[55] and one-sided Fisher's exact tests.

**Reporting summary**. Further information on research design is available in the Nature Research Reporting Summary linked to this article.

## Data availability

All data used in this study are publicly available. CBTTC molecular data are available through the public project on the Kids First Data Resource Portal and Cavatica (https://cbttc.org/) and through the PedCBioPortal (https://pedcbioportal.org/). PCAWG data are available at the ICGC Data Portal (https://dcc.icgc.org/pcawg). TCGA expression and SNP array-based CNA data are available from the Broad Institute Firehose pipeline (http://gdac.broadinstitute.org/). Access to controlled data from CBTTC or ICGC may be obtained from the respective consortiums through data use agreements. The remaining data are available within the Article, Supplementary Information, or available from the authors upon request. Source data are provided with this paper.

## Code availability

Source code in R for the linear modeling integrating SSV with expression data, with example data files, is available at Github https://github.com/chadcreighton/SV-expression_integration.

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

## Acknowledgements

This work was made possible through the resources and datasets made available by the Children's Brain Tumor Tissue Consortium (CBTTC). We thank the patients and their families for their participation in the CBTTC project. We also thank the patients and their families who participated in the individual ICGC and TCGA projects. This work was supported by National Institutes of Health (NIH) grant P30CA125123 (C.J.C.).

## Author contributions

Conceptualization: C.J.C.; methodology: C.J.C., F.C., Y.Z.; investigation: C.J.C., F.C., Y.Z., L.A.D., M.E.S.; formal analysis: C.J.C., F.C., Y.Z., L.A.D.; data curation: C.J.C.; visualization; C.J.C., F.C.; writing: C.J.C.; manuscript review: F.C., Y.Z., L.A.D., M.E.S.; supervision: C.J.C.

## Competing interests

The authors declare no competing interests.
