## [Peer Review File · Nature Communications]

REVIEWER COMMENTS

Reviewer #1 (Remarks to the Author):

In this manuscript the authors have analyzed a cohort of 854 pediatric brain tumors from the Childrens Brain Tumor Tissue Consortium (CBTTC). They have used the available whole genome sequencing and RNA sequencing data of this cohort to analyze the effect of somatic structural variations on genes near the breakpoints. Recently, the authors performed a very similar analysis on more than 2000 adult cancers from 38 different tumor types. In that study they reported that structural variations recurrently alter the expression of hundreds of genes, including several known oncogenes and tumor suppressor genes. As that study did not contain any pediatric brain tumors, they now repeated their analyses on a large cohort of pediatric brain tumors from the CBTTC consortium.

General comments:

The CBTTC cohort of pediatric brain tumors that has been analyzed in this manuscript includes 854 tumor samples from 759 individuals. 170/854 tumor samples from 75 patients even represent multiple tumors taken from the same patient, including both multiple primary tumors and recurrences of original tumors. The tumor samples span at least 33 different tumor types, all based on the histological diagnosis provided by the CBTTC database. However, there is no information at all provided how these histological diagnoses were made and moreover, and that is immediately the major shortcoming of this paper, no attempt at all has been made by the authors to classify these tumors molecularly. Pediatric brain tumors are very heterogeneous and many distinct types of brain tumors exist. However, it is well known and commonly accepted that a classification based on only histology is not sufficient. There are many different brain tumor types that may look histologically very similar, but based on molecular data really represent different entities. The other way is also true, many molecularly defined brain tumor types cannot be recognized by histology only as they may display a variety of different histologies. Therefore, when using histology only, many misdiagnoses occur. Moreover, it is also well-known now that several histologically AND molecularly defined entities are not a single disease but include multiple distinct subgroups that can be genetically totally different and often also have a totally different clinical behavior. These entities, including low grade gliomas, high grade gliomas, medulloblastomas, ependymomas, and ATRTs, have all been grouped together now, and no attempts have been made to classify them properly at the molecular level and further subgroup them. Also, CNS-PNETs are no longer seen as a distinct entity but may represent many different distinct entities and should thus not be grouped together. The authors even left in a class of tumors, designated as 'others' for which it is totally unclear what these are. If not clear, they should be removed. Also, what are the metastatic secondary tumors? Totally unclear what kind of entity they represent. Furthermore, they included some entities, such as neuroblastoma, that does not occur in the brain. Unclear if this is a misdiagnosis or a mislabeling and is not a brain tumor. If such cases cannot be clarified, they should be removed from the analyses.

Also, the short names the authors have used for all the different histologies are in many cases quite uncommon (eg PLGG, PHBB, EPMT, etc.) and should be more in line with what is commonly used in the literature.

The authors have used the entire cohort of 854 tumors to analyze structural variations and how they affect the expression of neighbouring genes. However, at several places they also analyzed subsets of these tumors, but numbers are often unclear, and also the rationale behind some of these subsets are unclear. A couple examples:

At page 6 they say: 'A subset of CBTTTC samples represented multiple tumors taken from the same patient, involving 170 tumor samples from 75 patients in total.'

But then later on at page 6 they say: 'Taking an example set of 36 cancer-associated genes, we examined somatic events, including SSV breakpoints occurring within 100kb of any genes, across 139 samples from 60 patients for which multiple tumor samples were profiled'

And at page 11, it is stated that: 'Of the 854 CBTTTC tumor sample profiles, 174 were from progressive or recurrent tumors and 633 were from initial tumors.'

Other major comments:

- Figure 1a: number of SSVs vary widely per tumor entity. Also quite some variation seen in some tumor entities, most likely because they represent totally different entities. Eg the PLGG and PHGG, but also within the ATRTs. As mentioned earlier, it would have been better if the authors had classified and subgrouped these entities better.

- Figure 1B: Why are the SSV gene associations total numbers different between the pie charts on the left vs the right?

- Page 6: 'Furthermore, in a global analysis of SSV breakpoint patterns across the 75 patients (Supplementary Figure 1d), inter-profile correlations between different tumors from the same patient ranged from very high to very low, low to the degree of suggesting little relationship between the samples. These results lent support to our treating each tumor sample as a separate disease entity in the downstream analyses. ' Unclear what they mean with separate disease entity? Or do they mean that they have treated each tumor sample as a separate sample regardless whether they came from the patient? But as they show there are also several samples that show a high degree of relationship as they also stated themselves: 'revealed both concordant and discordant patterns among samples from the same patient'. However, it is unclear whether they refer here then to multiple primary tumors or primary-relapse pairs from the same patient.

- However, at a later stage they say again: 'We found that different tumors from the same patient tended to demonstrate extensive molecular heterogeneity among them.' Again, unclear whether they have compared here multiple primary tumors from the same patient or primary relapses. I can understand that this is obvious when analyzing distinct tumors from the same patient, but to what extent was this also true when comparing the primary tumors with the recurrences? It does not make sense to group these all together in these analyses.

- Figure 1C: does not make sense mixing up multiple primary tumors from the same patient or primary relapse pairs in the same figure, rather display them then separate as two different panels.

- Page 8: 'A fraction of the positive correlations appeared to reflect SSV-mediated disruption of topologically associated domains'. However, this is all based on the TAD calling of the lung fibroblast IMR90 cell line, but it is totally unclear if these TADs are the same in every other tumor tissue they have analyzed in this cohort. It is rather more and more likely that tumors have tissue specific TADs.

- Suppl Table 3: in this table the authors provide a list of genes for which the expression is affected by SSVs in specific diseases only. It would be helpful if the authors can then also provide the actual expression data for these genes. Also, when checking these genes in for instance other publicly available databases, I noticed that many genes listed here are not at all expressed in entities where they should have been upregulated. How can the authors explain this?

- Fusion calling: despite using different algorithms for the fusion calling in the different entities, some of the well-known fusions in pediatric brain tumors, such as C11orf95-RELA fusions in supratentorial ependymoma, are not even included in the tables. This raises then the question how many more may have been missed? Also, they should have included in the tables in which tumor and in which entity each fusion was detected.

- A proper molecular classification of all tumor samples becomes again relevant when the authors report histone 3 mutations in different entities, including ATRT, MB, and PNET, while it is well known that these mutations are almost exclusively only found in HGGs. They should thus have checked whether these ATRTs, PNETs, and MBs are truly what they should be or are maybe all misdiagnosed HGGs?

- Overall mutation calling: in Suppl Table 1 the authors list per sample how many mutations (SNVs and indels?) they identified in that specific sample. Unclear whether this includes only coding mutations or genome wide all somatic mutations. That should be indicated. Furthermore, it is known that the number of mutations called per tumor in the CBTTC database are way higher than has been reported in the literature for the same entities. However, it remains unclear whether the authors have directly taken over the mutation calls from the CBTTC database or have called them again using their own pipelines. If the mutation calls are still way higher than reported before, how do the authors explain that then? Furthermore, in Suppl Table where the authors list all the mutations identified in all the samples, they should indicate not only the patient sample, but also the entity in this table, so that one can see tumor specific mutations. Finally, how come there no PTCH1 or SUFU mutations reported in this table as these are quite common mutations found in medulloblastoma? This makes the data all very unreliable.

Overall, my conclusion is that this is a very poor study and does not provide any further insights in any of the different pediatric brain tumor types included. The fact that breakpoints affect the expression of nearby genes and especially when it involves gene fusions is not new. When properly done, a thorough analysis of all the breakpoints in pediatric brain tumors and how that affects the gene expression of neighbouring in specific well-defined tumor types might be useful, but not in the way as has been done in this manuscript.

Other minor comments:

- Some gene names in the supplementary tables have been converted in dates

Reviewer #2 (Remarks to the Author):

This ambitious study about SSVs and their impact on gene expression changes in pediatric brain tumors is generally well written and carefully crafted. As clearly pointed out in the introduction, similar analyses have recently been performed on multiple adult cancers in a very comprehensive manner. The main novelty of the current study is thus simply that pediatric brain tumors have not been carefully analysed in the same way. This seems important enough to warrant publication in my opinion, in particular given the ambitious nature of the work and the large size of the CBTTC cohort. The low mutational burden typical of pediatric tumours may also make SV-based analyses particularly interesting. A general criticism, which does not necessarily have to be addressed but is nevertheless worth thinking about to maximise the appeal, is that it can be slightly exhausting to penetrate all the results. The actual biological novelty can become a bit lost among all the numbers, and I believe it would be beneficial to sometimes more clearly relate the results to the prior literature.

General comments regarding the core CNA-to-expression analysis:

Linear regression models are tricky in this context: tumor expression data tends to be very non-normal (in particular when it comes to lowly expressed genes) and model assumptions are certain to be violated quite frequently. Here, the relatively large number positive results obtained in permuted data can be noted (74 perm. vs. 295 observed - higher than it should be at 10% FDR, if model assumptions were completely correct). That makes the permutations important, as they may provide a more realistic point of reference (i.e. a better indication of the actual FDR). Important details are missing, however. Was the SV data permuted across the whole cohort, or within cancers types? If whole cohort, I would strongly suggest to permute within cancer types as this should provide a more realistic background distribution: otherwise, the "tumor-type-to-expression" structure, which is accounted for in the regression, is disrupted when permuting, which does not really make sense. It would be useful to clearly present the permutation data together with the observed data, either by adding to main figure 2b or at least discussing it conjunction with the main results (presently, this data is simply presented as "Widespread significant patterns were reflected in permutation testing..." toward the end, which doesn't say anything quantitatively). These aspects

are very central imo, forming the foundation of many numbers presented throughout the study (e.g. "100s of genes") .

What method was used to compute FDRs?

In relation to the normality assumption, at what levels are the significantly associated genes expressed, compared to other genes? A preference for lowly expressed genes could point to a bias toward normality-violating genes.

Other comments:

Pg 5 row 103, Fig S1a: "patterns markedly differing from those of adult brain tumors from TCGA". The adult reference is just a single cancer type (glioma), so perhaps hard to make general conclusions? Some of the pediatric cancers may in fact have profiles that are similar to LGG.

Fig. S1b: the large number of CNA amplifications associated with e.g. translocations seem surprising, as earlier studies have shown that a large fraction of them are specifically tandem duplications, as expected. Perhaps an overview of the absolute number of the different SVs types - or their relative abundances - could be useful. Translocations are often less common than e.g. tandem duplications, but S1b makes me think it is very different here? Some tandem duplications lead to gene fusions (e.g. KIAA1549-BRAF mentioned below) - I assume these are still classified as tandem duplications?

Fig. 1b: at least to me, this figure is difficult to understand and does not illustrate to points made in a very obvious way.

The text regarding fusions fails to even mention the extent to which fusions were in frame or not. If selection is generally absent, i.e. these are just random events or due to technical noise, about 1/3 should be in frame. In particular, it would make sense to mention whether the highlighted cases (e.g. BRAF) are in frame.

Is it BRAF-KIAA1549 or KIAA1549-BRAF?

Fusion findings - and the same can be said generally for many findings in the study - would benefit from being presented while briefly referring to the existing literature. E.g. KIAA1549-BRAF is known and described since before, but the reader has no way of knowing whether this is a novel finding or not. Is the frequency consistent with earlier reports? This particular case arises due to a local tandem duplication. This is also quite essential information and rather different from e.g. an inter chromosomal translocation, but not mentioned. Naturally, there needs to be a balance due to limited space, but I believe some of the results need to be put into context a bit more with respect to prior findings.

Again in Fig. 4b, in the case of classical gene fusions it would be important to consider the reading frame (or clarify that consideration has been taken)

Fig 6, SVs vs survival: is the analysis performed on the complete cohort? Different cancer types represent completely different diseases and may thus have wildly different survival properties. In principle, results like this can be obtained due to a single cancer type having unusually high SV burden, and also short survival - which does not necessarily mean much biologically speaking. At least some more details are needed in the text such that the results can be properly interpreted. Also, references to the prior literature regarding SV or CNA burden vs survival would be useful such that it is more clear to what extents novel insights are being gained (again).

Row 281: this another examples where a literature reference might be useful - is it known that high CNA or SV burden is associated with worse survival?

SSV burden vs. gene expression: also here, a linear model can be problematic, and results from permutations (within cancer types) would be useful to assess the true FDR.

Row 325: was the mutual exclusivity with P53 significant (Fisher's exact test)?

Methods, Row 506: please explain the weighted method more clearly.

Minor:

Although the manuscript is generally well written, I believe some more careful proofreading is needed to ensure all sentences are completely clear and grammatically correct. Examples:

Grammar row 137: "These results lent support to our treating each tumor sample as a separate disease entity in the downstream analyses."

Row 208, missing word.

Reviewer #4 (Remarks to the Author):

Somatic structural variants (SSVs) are an understudied class of variation in cancer, and their global impact on gene expression in many tumor types – including pediatric brain tumors is largely unknown. The authors combine cancer genome and RNA sequencing data from 854 tumors of thirty different childhood cancers to uncover associations between somatic SVs and transcriptomic changes. They describe different mechanisms leading to expression differences including gene fusions, altered cis-regulation events, gene disruption events, and altered gene dosage in this regard. The authors also associate high overall SSV burden with gene aberrations. From a methodological perspective, I did not notice any innovations in this study. The approaches described here have been used in prior applications of the same authors published in *Nat Common* and *Cell Reports*, which in many ways read quite similar to the current paper – with the exception that the focus of this manuscript is on childhood tumors which in prior studies relating SSVs to transcriptomic changes have received somewhat less attention. With respect to the choice of tumour types, I thus acknowledge that I do see a need for a study to search for tumor-driving SSVs, especially since this class of variation has received less attention in prior studies than copy-number changes or SNVs.

While the study is largely carried out with previously devised methodology, I have a number of issues with the interpretation of the data. The authors seem to assume throughout the manuscript that correlation (association) can be equated with causality, which is worrisome. But clearly, just because two things (ie the presence of an SSV and a gene expression measurement) correlate does not necessarily mean that one causes the other (in this case that SSVs deregulate gene expression). Prior studies have used a number of ways to address the issue of causation vs correlation with respect to SSVs and gene expression dysregulation (including by searching for evidence of allele-specific expression, by investigating enhancer elements, or by performing 4C-seq based validations; see e.g. PMID: 27869826; 32632335, PMID: 28726821). I found the lack of appreciation of these important aspects in the current manuscript disappointing. If the authors get the chance to submit a revised manuscript I would at least expect that they acknowledge this key issue, and demonstrate by using 4C-seq validations and/or haplotyping to generate allele-specific data further

support/verification of their conclusions that genes are “deregulated by somatic rearrangements” (as currently stated e.g. in the title).

Our knowledge on the location of cis-regulatory elements and enhancers along with genome of distinct tissues and cell types has vastly grown over the last decade. Can the authors make use of this resource of information, to evaluate whether in cases where intergenic SSVs correlate with gene expression this is associated with the specific juxtaposition (enrichment) of cis-regulatory elements / enhancers near those genes that become overexpressed?

The number of SSVs that the authors report as correlating with gene expressions seems (at least to me) unexpectedly high, which is worrisome. Could this be a result of the methodology used, which might be too permissive? The authors should provide QQ plots. What is the inflation factor λ ?

Our responses and comments are marked in red italics.

Reviewer #1 comments:

In this manuscript the authors have analyzed a cohort of 854 pediatric brain tumors from the Childrens Brain Tumor Tissue Consortium (CBTTC). They have used the available whole genome sequencing and RNA sequencing data of this cohort to analyze the effect of somatic structural variations on genes near the breakpoints. Recently, the authors performed a very similar analysis on more than 2000 adult cancers from 38 different tumor types. In that study they reported that structural variations recurrently alter the expression of hundreds of genes, including several known oncogenes and tumor suppressor genes. As that study did not contain any pediatric brain tumors, they now repeated their analyses on a large cohort of pediatric brain tumors from the CBTTC consortium.

We thank the reviewer for evaluating our work.

The CBTTC cohort of pediatric brain tumors that has been analyzed in this manuscript includes 854 tumor samples from 759 individuals. 170/854 tumor samples from 75 patients even represent multiple tumors taken from the same patient, including both multiple primary tumors and recurrences of original tumors. The tumor samples span at least 33 different tumor types, all based on the histological diagnosis provided by the CBTTC database. However, there is no information at all provided how these histological diagnoses were made and moreover, and that is immediately the major shortcoming of this paper, no attempt at all has been made by the authors to classify these tumors molecularly.

We explore the molecular classification question below under the next comment. Independent pathology review at the CBTTC centralized biorepository confirmed the histologic designations originally provided by the individual CBTTC member institutions contributing samples. We have noted this in the revised manuscript. (We have been periodically attending the monthly CBTTC virtual office hours, which provides users of the data the opportunity to ask questions, such as the above.) The pathology-based assessments appear to have carefully collated and could be considered accurate. Certainly, some tumors may be challenging to characterize, e.g., by showing molecular features that would be atypical for its associated histologic type.

In the manuscript revision, we have made a slight but notable change to the figures: We have renamed the previous “tumor type” color bars to “histologic type,” based on the assigned histology. For some, the “true” type of a pediatric brain tumor may involve a pathology-based assessment combined with molecular feature annotation. For our study’s purposes, however, we think it best to keep the molecular-based view of the CBTTC tumors separate from the pathology-based view. In our figures, certain molecular features may align well with specific histologic types as expected, but with a few exceptions (e.g., a few non-PLGG tumors could harbor KIAA1549-BRAF fusions, or a couple of non-EPMT tumors could harbor C11orf95-RELA fusions). We wish to show such discrepancies plainly in full view. Such discrepancies would not necessarily mean that the pathologist made a mistake, even though some readers can make their conclusions.

Pediatric brain tumors are very heterogeneous and many distinct types of brain tumors exist. However, it is well known and commonly accepted that a classification based on only histology is not sufficient. There are many different brain tumor types that may look histologically very similar, but based on

molecular data really represent different entities. The other way is also true, many molecularly defined brain tumor types cannot be recognized by histology only as they may display a variety of different histologies. Therefore, when using histology only, many misdiagnoses occur.

The histologic diagnosis and the molecular profile of a tumor sample would represent two distinct levels of information. In general, the molecular-based view and pathology-based view would be largely concordant, though with some discrepancies. Some tumors may be challenging to classify, either pathologically or molecularly. We would see no clear path to obtaining a new set of tumor designations, one that supersedes the well-established pathology-based designations and yet broadly accepted as superior to the pathology. The molecular characterization of a tumor can vary considerably, depending, for example, on what molecular features are considered and what algorithms and statistical cut points are used. It would not be trivial for us to take the CBTTTC molecular data and use them to call some CBTTTC samples as being “misdiagnosed.” The CBTTTC, as well as some pathologists, would likely take issue with such an approach.

In the manuscript revision, we explore the molecular subtype of the CBTTTC samples, using unsupervised hierarchical clustering of RNA-seq data from the 854 tumors (using the top 2000 most variable genes). The figure below is included in the revision as Supplementary Figure 3a. As expected, the tumors broadly segregated according to histologic type. There is some variability in terms of histology associations. Individual tumors that do not associate with their given histology can do so for many reasons, including their representing some unique and distinguishing biology.

Using the above results, we could broadly classify the CBTTTC tumors according to a molecular subtype. We use a designation of “-e” for “enriched,” to distinguish from the histology-based type. For example, “phgg-e.1” would represent a “phgg-enriched” cluster (with two such clusters identified, hence we distinguish them by “1” versus “2”). For the sake of information, we have added these molecular subtype designations in the sample-level information table in Data File S1. Using a different subtyping algorithm, or splitting the above clustering tree differently, would yield somewhat different molecular designations. Below, we explore these molecular designations a bit further regarding how these might influence our top SSV-altered genes.

In defining genes altered in the presence of nearby SSV breakpoints, we included the histologic type of the tumor as one of the covariates in the model. This covariate would help correct genes that might correlate with the tumor histology more so than the nearby breakpoint pattern. In the new Supplementary Figure

3b, we consider the use of molecular subtype as a covariate. This figure (also shown below) represents the numbers of significant genes ($p < 0.001$), showing a correlation between expression and SSVs occurring within 1Mb of the gene (using our distance metric method). Different linear regression models are considered here, as indicated, including using our molecular-based subtype (from the figure above) instead of the histologic designation.

Using molecular subtype instead of histologic type, we get very similar results to those reported in the main manuscript. Correcting for the tumor type does slightly decrease the number of significant genes. Most of the 324 genes significant at $FDR < 10\%$ using the histologic type+CNA model (from main Figure 2b), were also significant in the alternate molecular subtype+CNA model, with just 11 genes not being significant ($p > 0.05$) for the model using molecular subtype (as shown below). None of the genes that were not significant in the other model were recognizable as cancer-associated genes and had not been highlighted in the paper. Considering the alternate model would not be getting us closer to the true biology involving SSV-expression associations. Rather, any differences observed between statistical models likely have more to do with statistical noise and cutoff choice than with one model being more correct.

To summarize, CBTT pathological review involved two independent steps, one review being at the member institution and the other at the CBTT centralized repository. The pathological designations should, therefore, be considered rather accurate. The molecular subtype represents a different view of the data from that the pathology-based designation. We recognize that some tumors may be challenging to classify, either pathologically or molecularly. While we used pathologic type as a covariate in our models, this variable was not a primary driver in defining SSV-expression associations. By RNA-seq

analysis, the tumors broadly segregated according to histologic type. Using a molecular subtype designation instead of the pathological type as a covariate in our statistical models did not yield substantially different results from before. For the main manuscript, we have therefore used our initial results based on the pathology-based annotations. For our study's purposes, using an alternate tumor type annotation in our models would not necessarily be considered demonstratively better as opposed to merely different (with many of these differences likely being attributable to statistical noise).

Moreover, it is also well-known now that several histologically AND molecularly defined entities are not a single disease but include multiple distinct subgroups that can be genetically totally different and often also have a totally different clinical behavior. These entities, including low grade gliomas, high grade gliomas, medulloblastomas, ependymomas, and ATRTs, have all been grouped together now, and no attempts have been made to classify them properly at the molecular level and further subgroup them. Also, CNS-PNETs are no longer seen as a distinct entity but may represent many different distinct entities and should thus not be grouped together.

Our study recognizes pediatric brain cancer as a heterogeneous disease representing numerous distinct histologic types and associated molecular features. This heterogeneity would include the molecular subtype distinctions noted above. A number of these previously-defined molecular subtypes have been defined from methylation arrays, although CBTTTC data do not include DNA methylation. For the purposes of defining SSV-expression associations, further subdividing tumors by molecular subtype in addition to histologic type would be more granular than what would be needed for the purposes of our pan-cancer SSV study. When we compare all the samples with each other in a global molecular analysis (e.g., in the above hierarchical clustering), the most-represented histologic types—including the low grade gliomas, high grade gliomas, medulloblastomas, ependymomas, and ATRTs—all form fairly homogenous groups. Our approach would be consistent with that of other studies by other groups. For example, in the Northcott et al. medulloblastoma study (PMID: 28726821), the medulloblastoma molecular subtype was not used in the SSV-expression modeling by CESAM method. Likewise, for our own study of adult cancers and SSVs (PMID: 32024823), carried out as part of the PCAWG consortium, we did not further stratify the cancers according to molecular subtype (e.g., the breast cancer intrinsic subtypes), but used tissue-of-origin as the covariate in our modeling.

Our pan-cancer approach, to identify patterns cutting across histologic types of pediatric brain tumors, is well-suited to the CBTTTC datasets. As described above, we include the histologic type of the tumor as one of the covariates in our statistical models defining genes altered in the presence of nearby SSV breakpoints. However, this covariate would not be something that is primarily driving the SSV-expression associations reported. Variations on annotating the tumor type by molecular rather than pathology information would not significantly alter the findings. In our results highlighting specific genes and samples, we take care to show the associated histologies that underlie the patterns involved. Regarding the PNETs, there are too few of these (~13 samples) for further re-classification to make any meaningful difference to our overall results and findings. It may well be the case that PNETs represent different distinct entities. However, as the CBTTTC-affiliated pathologists have classified these tumors as PNETs, we wish to respect this and use the PNET designation in the figures. For us to do otherwise would likely raise concerns regarding our consistency.

The authors even left in a class of tumors, designated as ‘others’ for which it is totally unclear what these are. If not clear, they should be removed. Also, what are the metastatic secondary tumors? Totally unclear what kind of entity they represent. Furthermore, they included some entities, such as neuroblastoma, that does not occur in the brain. Unclear if this is a misdiagnosis or a mislabeling and is not a brain tumor. If such cases cannot be clarified, they should be removed from the analyses.

In the revised Data File S1 (sample-level information), we have added another data field, entitled “ADDITIONAL PATH NOTES – OTHER” (column G). For each of the 41 tumors classified as “Other” or “Metastatic secondary tumors” (column F), we include any relevant notes from the pathology reports sent to CBTTTC from the contributing tissue sites. The “Other” tumors did not fit within the CBTTTC’s set of pre-defined histologic categories. About seven tumors did not have a pathology report. The full pathology reports are available from CBTTTC, though access may need to be requested.

As we had explicitly noted in the manuscript, some tumor types represented in CBTTTC—including Ewing’s sarcoma (ES, n=6), germinoma (GMN, n=4), Langerhans cell histiocytosis (LCH, n=4), malignant peripheral nerve sheath tumor (MPNST, n=3), and neuroblastoma (NBL, n=2)—originate from cell types not specific to the brain, even if the CBTTTC tumors were obtained from the brain region. A subset of tumors were either difficult to classify in terms of histology or did not fit into the pre-defined categories and so were designated by CBTTTC as “other” (n=36). An additional five tumors were “metastatic secondary tumors” involving pediatric brain tumor patients but presumably not aligning within one of the other histologic types. The above categories are important for us to keep in mind, although these would represent a minority of the CBTTTC 854 tumor set.

The CBTTTC’s regulatory for CNS tumors intentionally allowed for the broad collection of “abnormal cell growth,” not necessarily specific to brain-specific cell types. For our study, we decided to include all 854 CBTTTC samples in the main analysis. All data that were available from CBTTTC, therefore, formed the basis of our study. We reasoned that these anomalous tumors would provide useful information that may help compare or contrast the patterns found in tumors originating from brain cell types. Where an altered gene regulation association involves brain and non-brain cell types, the tumors not originating from brain cells would provide additional power for us to detect such associations. By nature, our study is not one that examines a uniform set of tumors of the same histology or class. Instead, we take advantage of the truly diverse set of tumors represented in our study, involving uniform sample processing and data generation on the same platforms.

In the new Supplementary Figure 3c, we consider a scenario where only CBTTTC tumors originating from brain cells are used in the analysis to define SSV-altered genes. Removing the above tumors in question (ES, GMN, LCH, MPNST, NBL, metastatic secondary tumors, and Not Otherwise Specified, n=61), 793 tumors remained out of the full set of 854. We ran the histologic type + CNA model (1Mb region) on the 793 tumor subset. The chart shown below represents the numbers of significant genes ($p < 0.001$), showing a correlation between expression and SNVs occurring within 1Mb of the gene (using our distance metric method), for both 854 and 793 tumor sets.

When removing non-brain cell-derived tumors originating from non-brain cells from the analysis, we get very similar results to those reported in the main manuscript. Using the smaller set of tumors yields similar numbers of significant genes observed when using the full CBTTTC tumor set. Most of the 324 genes significant at FDR<10% using the histologic type+CNA model and the full 854-tumor set (from main Figure 2b) were also significant in the alternate analysis using the 793 tumor subset, with just 31 genes not being significant ($p>0.05$) for the 793 tumor subset (as shown below). None of the genes that were not significant in the other model were recognizable as cancer-associated genes and had not been highlighted in the paper. Many of these left-out genes could still represent false negatives, however.

Also, the short names the authors have used for all the different histologies are in many cases quite uncommon (eg PLGG, PHBB, EPMT, etc.) and should be more in line what is commonly used in the literature.

Our study uses histology abbreviations as provided by CBTTTC. This practice allows for consistency across different studies that use CBTTTC data. The abbreviations are also well-suited to the shorthand needed for utilizing space in the figures most effectively. Our use of CBTTTC's abbreviations is similar to what has been done previously in studies using TCGA data.

The authors have used the entire cohort of 854 tumors to analyze structural variations and how they affect the expression of neighbouring genes. However, at several places they also analyzed subsets of these

tumors, but numbers are often unclear, and also the rationale behind some of these subsets are unclear. A couple examples:

In general, if a subset of tumors is represented in a figure, it is usually the set of tumors for which at least one of the molecular features represented showed alterations. For example, the pathways figure (Figure 5c, formerly 4c) shows just the altered tumors for a given pathway. In other cases, the analysis focused on a specific subset of tumors, such as the tumors involving multiple tumor samples being taken from the same patient (Figure 1c) or the analyses focusing on progressive or recurrent samples (Figure 6, formerly Figure 5).

At page 6 they say: ‘A subset of CBTTC samples represented multiple tumors taken from the same patient, involving 170 tumor samples from 75 patients in total.’ But then later on at page 6 they say: ‘Taking an example set of 36 cancer-associated genes, we examined somatic events, including SSV breakpoints occurring within 100kb of any genes, across 139 samples from 60 patients for which multiple tumor samples were profiled’

We can see how this might have been a bit confusing. We took 139 tumors from the full 170 that involved multiple tumors taken from the same patient. The 139-tumor subset is the tumors for which at least one of the 36 genes associated with an SSV breakpoint. In the revision, we have clarified this point in the figure legend and corresponding section of Results.

And at page 11, it is stated that: ‘Of the 854 CBTTC tumor sample profiles, 174 were from progressive or recurrent tumors and 633 were from initial tumors.’

The above is correct, as stated: $174+633=807$, with the remaining tumors being either second malignancy or not having the tumor status available (Data File S1). As described in the Results section involving main Figure 6, we applied our analytical approach separately to the full subset of initial tumors ($n=633$) and the full subset ($n=174$) of recurrent or progressive tumors. This analysis allowed us to study SSV-mediated alterations specifically involved with more advanced disease.

Other major comments:

- Figure 1a: number of SSVs vary widely per tumor entity. Also quite some variation seen in some tumor entities, most likely because they represent totally different entities. Eg the PLGG and PHGG, but also within the ATRTs. As mentioned earlier, it would have been better if the authors had classified and subgrouped these entities better.

Our study recognizes pediatric brain cancer as a heterogeneous disease representing numerous distinct histologic types and associated molecular features. Any molecular stratification that we might apply to the CBTTC cohort would also show a wide range of detected SSVs within each grouping. This wide variation within histology would be analogous to what PCAWG and other studies have observed in adult cancers of various types (e.g., breast vs. ovarian vs. colon, etc.). Above, we address the issues involved with using a molecular-based approach to categorize the tumors, in place of the histology designations provided by CBTTC.

- Figure 1B: Why are the SSV gene associations total numbers different between the pie charts on the left vs the right?

Figure 1b depicts an overall association of CNAs with genomic rearrangement events. This is a phenomenon that we have previously reported for adult cancers, and which we report here for the pediatric brain cancers. In both Venn diagrams, the same set of all SSV-gene associations involving an SSV breakpoint falling within 1Mb of gene start site for a given sample is represented (1171040 total associations). As noted in the Figure legend, the difference between the two Venn diagrams is that the left one shows the overlap with high-level gene amplification events, and the right one shows the overlap with deep gene deletion events. The total geneXsample associations involving nearby SSV breakpoints is the same in both diagrams. Still, the set of overlapping geneXsample events involving amplification or deletion, respectively, will differ, so the remaining events (total minus the overlap) will also differ.

- Page 6: ‘Furthermore, in a global analysis of SSV breakpoint patterns across the 75 patients (Supplementary Figure 1d), inter-profile correlations between different tumors from the same patient ranged from very high to very low, low to the degree of suggesting little relationship between the samples. These results lent support to our treating each tumor sample as a separate disease entity in the downstream analyses. ‘Unclear what they mean with separate disease entity? Or do they mean that they have treated each tumor sample as a separate sample regardless whether they came from the patient? But as they show there are also several samples that show a high degree of relationship as they also stated themselves: ‘revealed both concordant and discordant patterns among samples from the same patient’. However, it is unclear whether they refer here then to multiple primary tumors or primary-relapse pairs from the same patient.

In our analysis, given multiple tumors from the same patient (e.g., involving multiple initial tumors, or initial compared to recurrent or progressive tumors), we considered each tumor sample separately from the others. So, yes, we have treated each tumor sample as a separate sample regardless of whether they came from the same patient. We have added further clarification of the above to the manuscript revision. The instance of multiple tumors assigned to the same patient would include samples taken from a patient at different times, e.g., samples taken from an initial tumor and later from a progressive or recurrent tumor, or samples representing a second malignancy. In other cases (involving 13 patients), multiple initial tumors were profiled, often taken from different sites, as a patient may present with multiple initial tumors upon diagnosis.

Both main Figure 1c and Supplementary Figure 1d include recurrent or progressive tumors as well as initial tumors. For reference, below, we show Supplementary Figure 1d. The tumor status is indicated by a color bar along the bottom of each patient. The tumor status color bar denotes initial tumor, progressive, recurrence, or second malignancy. In the revised manuscript, we have noted the tumor status categories for the figure legends of main Figure 1c and Supplementary Figure 1d.

As shown in the above, while in many instances, we observed the global correlation between tumors from the same patients to be quite high, these correlation r -values are much less than 1. In many other instances, the correlation level is close to that of unrelated tumors from different patients.

Our decision to treat each tumor as a separate disease entity is one meant to balance some tradeoffs. On the one hand, we cannot assume that a somatic alteration in one patient's tumor would be present in the other tumors. On the other hand, we may observe events in multiple tumors from the same patients for some genes, which undoubtedly helps establish an association, even if this might apply to fewer patients than tumors. However, we are ultimately interested in repeated patterns, and so if these occur, for example, in multiple independent tumors from the same patient, such patterns remain of interest to us. An alternate approach of selecting only one tumor per patient for the analysis, or somehow averaging tumor profiles within patients, would have raised other issues and concerns.

- However, at a later stage they say again: 'We found that different tumors from the same patient tended to demonstrate extensive molecular heterogeneity among them.' Again, unclear whether they have compared here multiple primary tumors from the same patient or primary relapses. I can understand that this is obvious when analyzing distinct tumors from the same patient, but to what extent was this also true when comparing the primary tumors with the recurrences? It does not make sense to group these all together in these analyses.

Our statement includes both multiple primary tumors and primary relapses from the same patient, as indicated in the tumor status bars of Figure 1c and Supplementary Figure 1d. As shown above, we find that recurrent tumors also often show little relation to the initial tumor. Our observations would be consistent with those of previous studies. For example, in studies of medulloblastoma (Morrissy et al., Nature 529:351-7, 2016, PMID:26760213), genetic events in recurrent tumors exhibited a very poor overlap (<5%) with those in the matched initial tumors. In the Discussion section of the revised manuscript, we have added a reference to the above Morrissy et al. paper as being in line with our observations.

- Figure 1C: does not make sense mixing up multiple primary tumors from the same patient or primary relapse pairs in the same figure, rather display them then separate as two different panels.

Figure 1 is intended to show inter-tumoral heterogeneity that exists for a given patient. Therefore, the figure is patient-centric, allowing one to see, within a given patient, what genes are altered across all the tumors for that patient. Breaking up the single panels into multiple panels would considerably complicate one being able to track what is going on for a given patient, as this would require moving one's eyes repeatedly between multiple panels. The tumor status bar indicates which tumors are initial tumors and which are progressive or recurrent or second malignancy in the current figure. Our findings on inter-tumoral heterogeneity within patients would apply to both multiple primary tumors and primary relapses from the same patient.

- Page 8: 'A fraction of the positive correlations appeared to reflect SSV-mediated disruption of topologically associated domains'. However, this is all based on the TAD calling of the lung fibroblast IMR90 cell line, but it is totally unclear if these TADs are the same in every other tumor tissue they have analyzed in this cohort. It is rather more and more likely that tumors have tissue specific TADs.

Following our previous studies in adult tumors (Zhang et al., Cell Reports 2018, PMID: 29996110; Zhang et al., Genome Biology 2019, PMID: 31610796), we used published TAD data from the IMR90 cell line (Dixon et al., Nature 2012, PMID: 22495300). As noted elsewhere (Weischenfeldt et al., Nature Genetics 2017, PMID: 27869826; Rao et al., Cell 2014, PMID: 25497547), TADs are largely invariant across cell types. Our use of IMR90 cell line data to define TADs follows the same practice as other studies, including Weischenfeldt et al. and Northcott et al. (PMID: 28726821).

- Suppl Table 3: in this table the authors provide a list of genes for which the expression is affected by SSVs in specific diseases only. It would be helpful if the authors can then also provide the actual expression data for these genes. Also, when checking these genes in for instance other publicly available databases, I noticed that many genes listed here are not at all expressed in entities where they should have been upregulated. How can the authors explain this?

There might be a couple of ways to explain the reviewer's observations. Detection in RNA-seq would depend on the depth of sequencing, so perhaps not all pediatric tumor RNA-seq datasets would have as deep coverage like that of CBTTTC. We should also note that the SSV-associated patterns would involve a small fraction of cases for each gene. There may be cases where tumors not associated with SSVs would have zero expression, while SSV-impacted tumors have non-zero expression. But most of the tumors referred to in an external database likely do not involve SSVs for a given gene of interest. For example, TERT, a gene associated with SSV-mediated up-regulation in many different studies and cancer types, has very low average log2 expression across all samples, at 0.22. But for the tumors harboring a nearby SSV breakpoint, the expression is quite high, as shown in main Figure 3.

In the box plot below, we consider the set of genes significantly increased (FDR<10%) for any one of the 13 tumor types examined individually. For each tumor type, we plot the average expression (log2) of each significant gene, involving just the tumors harboring an SSV breakpoint within 1Mb for that cancer type. Box plot represents 5%, 25%, 50%, 75%, and 95%.

Noting the log₂ scale, we see moderate-to-high average expression for most genes. Although some genes may appear close to zero in the plot, there would be no genes here with zero or non-detectable expression (by definition, the expression must be higher for SSV-impacted tumors). Of course, for the non-SSV-associated tumors (not represented here), the average gene expression trends lower for the same genes.

- Fusion calling: despite using different algorithms for the fusion calling in the different entities, some of the well-known fusions in pediatric brain tumors, such as C11orf95-RELA fusions in supratentorial ependymoma, are not even included in the tables. This raises then the question how many more may have been missed? Also, they should have included in the tables in which tumor and in which entity each fusion was detected.

In fact, we had missed the C11orf95-RELA fusion in ependymoma! Based on the reviewer's comment, we went through our analysis to understand why this fusion had not been included. This effort resulted in a revised Figure 4 (shown below, formerly Figure 3) and Supplementary Figure 4 and Data File S4. The one change involving Figure 4c, for example, is the addition of the C11orf95-RELA fusion, involving 22 samples and 20 patients, including 20 EPMT tumors.

The reason we had previously missed the C11orf95-RELA fusion is that we had narrowed the space of fusion candidates for consideration to the set for which BOTH genes were represented in the gene-level expression dataset. However, the expression dataset provided to us by the PedCBioportal did not include C11orf95 for some reason. Therefore, we had removed the fusion calls by RNA-seq algorithm involving C11orf95 in the pre-filtering stage, before considering the overlap with our SSV-based results. For the revision, we have expanded the analysis to consider fusion candidates for which EITHER gene is represented in the expression dataset. This change allows us to include C11orf95-RELA as well as other fusions in our final results. Regarding C11orf95-RELA, we found that RELA gene was highly over-expressed in the fusion tumors, coinciding with SSV breakpoints detected in both C11orf95 and RELA. We are very thankful that the reviewer alerted us to this!

We can perhaps note that two of the tumors with C11orf95-RELA were not of the EPMT histologic type. We feel this would not be sufficient for us to conclude that the pathologists who first assessed these two tumors had made a mistake. Instead, it is conceivable that some tumors that associate with one group in terms of histology may associate with another group in terms of observed molecular features. We expect the tumor histology and the tumor molecular profile to coincide broadly, but this still allows for observed exceptions to the general rule.

- A proper molecular classification of all tumor samples becomes again relevant when the authors report histone 3 mutations in different entities, including ATRT, MB, and PNET, while it is well known that these mutations are almost exclusively only found in HGGs. They should thus have checked whether these ATRTs, PNETs, and MBs are truly what they should be or are maybe all misdiagnosed HGGs?

The H3F3A mutations have indeed been observed primarily in HGGs and DIPGs, but this would not exclude the possibility of these histone 3 genes also being mutated in other cancer types, though involving a very small fraction of cases. Based on the hierarchical clustering of the RNA-seq profiles (see above), we find that the three histone 3-associated PNET samples in Figure 8b associate with pHGG (consistent with the reviewer's point that PNETs may not truly represent a distinct disease entity). However, the ATRT and MB tumors did not associate with pHGG by molecular subtype. By hierarchical clustering, one ATRT tumor was assigned to a "mixed" cluster, but the other ATRT and the MB tumor were respectively assigned to "ATRT-e" and "MBL-e" clusters.

Regardless of what the molecular profile might suggest, even in light of our previous expectations of what ought to define the histologic types, we would be hesitant to change the CBTTTC-assigned histologic designations as represented in the figures based on molecular data. To do this would likely garner pushback from many readers. CBTTTC pathological review involved two independent steps, one review being at the member institution and the other at the CBTTTC centralized repository. The pathological designations should, therefore, be considered fairly accurate. As we explain above, it would not be a trivial thing to take the CBTTTC molecular data and use them to call some CBTTTC samples as being "misdiagnosed." For the study's purposes, it would represent somewhat circular thinking for us to start with the pathologic designations that then conclude in the paper that these designations must be wrong, based on some molecular feature. We think it best that our study presents the pathology-based view independently of the molecular-based view. If we start down a road of using the molecular data to "correct" the pathology-based assessments, this will lead to serious issues with consistency.

- Overall mutation calling: in Suppl Table 1 the authors list per sample how many mutations (SNVs and indels?) they identified in that specific sample. Unclear whether this includes only coding mutations or genome wide all somatic mutations. That should be indicated. Furthermore, it is known that the number of mutations called per tumor in the CBTTTC database are way higher than has been reported in the literature for the same entities. However, it remains unclear whether the authors have directly taken over the mutation calls from the CBTTTC database or have called them again using their own pipelines. If the mutation calls are still way higher than reported before, how do the authors explain that then?

The overall mutation count was provided directly by CBTTTC investigators and was not from our group. We used the "mutation count" field from the clinical data files provided by PedCBioPortal (<https://pedcbioportal.org/>). The mutation count comes from exome analysis, not whole genome analysis, and so we have made a note of this in the revised manuscript (e.g., in Supp Table 1, methods, and figure legends). (We have been periodically attending the monthly CBTTTC virtual office hours, which provides users of the data the opportunity to ask questions, such as the above.) The use of the mutation count data was for secondary analyses and not needed to derive our top SSV-associated genes. As provided by the CBTTTC, the exome mutation count data seems to align consistently with other molecular modalities we examined in our study, including SSV counts. It is not clear how these data might be inconsistent with reports in the literature. We should note that while there may be broad generalizations according to

tumor type, there can be a wide range represented within each tumor type, with individual tumors varying considerably from the average. If an individual tumor were to differ dramatically from its assigned group, according to a given measure such as numbers of mutations, this in and of itself should not be considered a misdiagnosis by CB TTC pathology.

Furthermore, in Suppl Table where the authors list all the mutations identified in all the samples, they should indicate not only the patient sample, but also the entity in this table, so that one can see tumor specific mutations. Finally, how come there no PTCH1 or SUFU mutations reported in this table as these are quite common mutations found in medulloblastoma? This makes the data all very unreliable.

There seems to be a misunderstanding here. Supplementary Table 5 does not provide mutations for all genes, but for genes in specific pathways, namely pathways involving SSV-altered genes (focusing primarily on the set of pathways highlighted in our previous paper, PMID: 28775315). This file would represent the data underlying main Figure 5, but not all mutations for all genes in the CB TTC cohort. We have clarified this point in the Supplementary Table 5 description. In particular, we did not highlight the Hedgehog pathway in main Figure 5, and this is the reason that PTCH1 and SUFU mutations do not appear in the Supplementary Table. PTCH1 truncation mutations are found in medulloblastoma in CB TTC cohort, involving 11 tumors and eight patients. One medulloblastoma tumor had a SUFU mutation. In the revision, we note the PTCH1/SUFU mutations in the legend of Supplementary Figure S5. The Supplementary Table has multiple tabs, including one that provides a sample versus mutation table (with histologic type), and another that provides the mutations in a standard MAF-style format. From this file, one should be able to get tumor-specific mutation information. One can also lookup mutations from other genes in the PedCBioPortal (<https://pedcbioportal.org/>).

Overall, my conclusion is that this is a very poor study and does not provide any further insights in any of the different pediatric brain tumor types included. The fact that breakpoints affect the expression of nearby genes and especially when it involves gene fusions is not new. When properly done, a thorough analysis of all the breakpoints in pediatric brain tumors and how that affects the gene expression of neighbouring in specific well-defined tumor types might be useful, but not in the way as has been done in this manuscript.

Our present study's pan-cancer approach would set it apart from other studies, whereby here we can identify patterns cutting across histologic types. As part of considerable efforts by the CB TTC, different histologic subgroups have undergone uniform sample collection and data generation, with combined WGS and expression data that would set these data apart from those of previous studies. Previous genomic studies involving pediatric brain tumors have explored the landscape of SSVs (notably Nature 555:321–327, 2018, PMID: 29489754), but without corresponding expression data to link the SSV breakpoints with the transcription of the nearby genes. A recent medulloblastoma study did feature both expression and WGS data on ~164 tumors (Nature 547:311-317, 2017, PMID: 28726821), which might be the only study comparable to ours, but representing only a single tumor type. Most of the significant genes that we found in our study involved cases spanning more than one histologic type. A few key genes, including EGFR and MYCN, were significant only in the pan-cancer analysis and not associated with any individual cancer type, highlighting the utility of our pan-cancer approach.

The fact that SSVs may alter gene regulation or form gene fusions is certainly well known. However, which specific genes may be altered in which cancer types remains to be explored, analogous to the host

of whole-exome studies uncovering different somatically mutated genes involved in different cancer types. As part of PCAWG, we systematically cataloged SSV-mediated cis-regulatory alterations across a wide range of adult cancers. In this present study, the CBTTTC data represented an opportunity to apply our integrative analysis methods to pediatric cancers, to examine SSV-mediated alterations in a way not previously done before in this cancer patient population. As expected, pediatric brain tumors would involve other genes with SSV-altered cis-regulation compared to adult cancers. We agree that future studies involving larger numbers of tumors for a given histologic type (as such data become available) could follow a similar approach as ours, to better define SSV-altered genes specific to that type.

Other minor comments:

- Some gene names in the supplementary tables have been converted in dates

We have gone through the supplementary tables and corrected this. All of our analyses relied on the gene Entrez id or Ensembl id, so the issue of Excel confusing some names for dates (e.g., "SEPT8") did not affect our analyses.

Reviewer #2 comments:

This ambitious study about SSVs and their impact on gene expression changes in pediatric brain tumors is generally well written and carefully crafted. As clearly pointed out in the introduction, similar analyses have recently been performed on multiple adult cancers in a very comprehensive manner. The main novelty of the current study is thus simply that pediatric brain tumors have not been carefully analysed in the same way. This seems important enough to warrant publication in my opinion, in particular given the ambitious nature of the work and the large size of the CBTTTC cohort. The low mutational burden typical of pediatric tumours may also make SV-based analyses particularly interesting. A general criticism, which does not necessarily have to be addressed but is nevertheless worth thinking about to maximise the appeal, is that it can be slightly exhausting to penetrate all the results. The actual biological novelty can become a bit lost among all the numbers, and I believe it would be beneficial to sometimes more clearly relate the results to the prior literature.

We thank the reviewer for evaluating our work. Our study is a survey of SSVs in pediatric brain tumors, and so it is broad by nature. It is known that, in principle, SSVs can alter gene expression in cancer, but for which genes and in which cancer types is something that remains to be flushed out. So, it would be the catalog itself of SSV-altered genes in the CBTTTC cohort that our study offers. Many of our observations regarding specific genes align with the results of previous studies in the literature. In these cases, we provide a literature reference as we state our observations.

General comments regarding the core CNA-to-expression analysis:

Linear regression models are tricky in this context: tumor expression data tends to be very non-normal (in particular when it comes to lowly expressed genes) and model assumptions are certain to be violated quite frequently. Here, the relatively large number positive results obtained in permuted data can be noted (74 perm. vs. 295 observed - higher than it should be at 10% FDR, if model assumptions were completely correct). That makes the permutations important, as they may provide a more realistic point of reference (i.e. a better indication of the actual FDR). Important details are missing, however. Was the SV data permuted across the whole cohort, or within cancers types? If whole cohort, I would strongly suggest to

permute within cancer types as this should provide a more realistic background distribution: otherwise, the “tumor-type-to-expression” structure, which is accounted for in the regression, is disrupted when permuting, which does not really make sense. It would be useful to clearly present the permutation data together with the observed data, either by adding to main figure 2b or at least discussing it conjunction with the main results (presently, this data is simply presented as “Widespread significant patterns were reflected in permutation testing...” toward the end, which doesn’t say anything quantitatively). These aspects are very central imo, forming the foundation of many numbers presented throughout the study (e.g. “100s of genes”).

There appear to be two main issues brought up in the above comments. One issue would be our use of linear modeling and whether expression data would align with model assumptions. The other issue would be that of using the permutation testing results to estimate FDR. We tackle these issues below, starting with the linear modeling issue, followed by the permutation testing issue. Our analytical approach as applied to the CBTTTC datasets is the same as what we have applied to adult cancers in our previous studies (Zhang et al., Cell Reports 2018, PMID: 29996110; Zhang et al., Genome Biology 2019, PMID: 31610796; Zhang et al., Nature Communications 2020, PMID: 32024823).

Linear modeling: *In all of the linear models performed in our study, we used appropriate data transformations to make the data align better with the model assumptions. In the Methods of the manuscript revision, we now state this explicitly. For example, gene expression data were log₂-transformed, a ubiquitous practice for linear model-based methods for differential expression, including SAM and limma. Similarly, relative SSV breakpoint distances from the gene in the breakpoint pattern matrix were also log-transformed. One of the main concerns with data not meeting model assumptions is that type II errors (i.e., false negatives) could result. In instances where model assumptions do not hold, this suggests that an alternative test might do better in detecting significant patterns, though every test has its limitations. Below, we further explore the impact of data transformations on results using our method.*

As an example of our effective use of data transformations for linear modeling, below, we examine genes BCAR4 (top) and IGF2 (bottom) in the PCAWG-TCGA datasets (from Zhang et al. Genome Biology, PMID: 31610796), which genes were highlighted in our previous paper as examples of significant genes. (Results were originally provided as part of <https://genomebiology.biomedcentral.com/articles/10.1186/s13059-019-1818-9#MOESM12>.) Gene expression levels corresponding to SSVs located in the genomic region 1Mb downstream to 1Mb upstream of the gene are shown, with absolute relative distance (upstream or downstream) represented on the x-axis. Each point represents a single case (closest SSV breakpoint represented for each case). On the left, both expression and relative distance are untransformed; in the middle, expression but not relative distance is log-transformed; on the right, both expression and relative distance are log-transformed. The figure below illustrates how using data transformations better defines linear relationships. All three plots would indicate some kind of linear relationship for each gene, but this is best revealed by the plots on the right, where the corresponding trend lines are steepest. Regarding relative distance, the log-transformation is doing exactly what we want it to do. The data points closest to the gene are expanded out, while data points closer to 1Mb are brought closer in; thus, the furthest data points are not being overly-weighted in the analysis.

The linear model indeed represents a theoretical distribution (which real data usually do not perfectly follow). Still, our models do a fairly good job of identifying significant patterns for many of our genes, given the appropriate data transformations. A real advantage of using a linear model is that it allows, in principle, easy integration of covariates, which is a critical aspect of our present study. Nonlinear models would not allow us to specify cancer type as a factor, for example, which would raise a host of concerns, including our desire to screen out genes that represent tissue-specific markers rather than SSV-altered genes. In general, nonparametric regression requires larger sample sizes than regression based on parametric models because the data must supply the model structure with the model estimates. Nonparametric regression models can become overfit either by including too many predictors or by using small smoothing parameters. This issue can make a big difference with special problems, such as small data sets or clumped distributions along predictor variables.

Permutation testing: There may be a point of confusion here. We did not use the permutation testing to estimate the FDR. We estimated FDR using the Storey and Tibshirani method (described further below). The permutation testing was done separately from the main analysis that used linear modeling for FDR estimation. The permutation testing is something we put forth as another indication there are widespread associations between SSV and expression, not due to chance expected or to the testing of multiple genes. We permuted the SSV data across the whole cohort. So, in 1000 tests of mixing up the data and the SSV-expression relationships, we do not get a set of nominally significant genes that would be anywhere close to what we get using the actual data. In each of our previous SSV studies, we included a similar permutation testing result in the Supplemental. The idea here was that some readers lean heavily toward computer science over statistics, and such readers may not always understand or accept classical statistical methods but seem to accept permutation-based approaches more readily. If we were to remove the permutation testing from the study entirely, it would not change any of the actual results or numbers in the other figures, since we did not use the permutation testing to estimate p-values or FDRs. For this reason, we have left the permutation testing result in the Supplemental, rather than moving it into a main figure, since this result is extraneous to the rest of the study.

As suggested by the reviewer's comments, there is likely no perfect way to carry out the permutation testing. We would see no way to carry out a pure computer science-based permutation approach, devoid of statistical modeling. Our approach requires the integration of covariates (tumor type and CNA) in addition to SSV and expression data, which is something only linear models can do effectively. Therefore, our permutation testing approach (mix up one dataset, then do the linear modeling) represents a hybrid of statistics and computer science. In shuffling one dataset, the other dataset and covariate structure remain intact. The sample size is limited, and so many permutations could result in a dataset that is not far removed from the original. Therefore, it would yield many nominally significant genes and inflate the FDR. Regarding the permutation result represented in Supplementary Figure 2a, we would consider the average of 74 $p < 0.001$ genes to be inflated and not representative of the true FDR, as per the above.

Permuting within tumor types, as suggested by the reviewer, actually makes the situation a bit worse since there is less possible divergence from the actual dataset (some tumor types having only 2-4 samples). We did try doing the permutation by mixing within tumor type instead of across the entire dataset, but the results were not satisfactory. This approach caused singular problems for many genes, so we had to limit the genes to those with $p < 0.20$ in the original linear model to complete the permutation testing. With the 4375 genes tested, of which 275 were significant ($p < 0.001$, linear model including CNA and cancer type), 1000 permutation tests yielded 32.4 $p < 0.001$ genes on average (the highest number for a single permutation being 51). In this instance, consistent with our other analyses, we see that mixing up the sample ordering results in far fewer nominally significant genes arising. The above would further demonstrate that the significant results from the actual dataset do not as a whole represent chance associations between SSV breakpoint patterns and gene expression, even if we are not using the actual permutation results to estimate FDRs.

What method was used to compute FDRs?

As we did in our previous SSV studies, we used the method of Storey and Tibshirani (PNAS 2003, PMID: 12883005) to estimate FDR. We did not use the permutation tests to estimate FDR, for the reasons described above. The Storey and Tibshirani method is widely used in gene expression profiling studies. The method uses the following formula for computing FDR for each gene: $[\text{nominal } p\text{-value}] \times [\text{total number of genes tested}] / [\text{total number of genes that were significant at the given } p\text{-value}]$. We have added this description of the method to the manuscript revision.

There would be several possible methods for estimating FDR, and in the end, these are but estimates. There are many more significant associations found in the actual dataset than what the chance expected would yield, whether we are using Storey and Tibshirani or some variation of permutation testing. We would not consider the FDRs as equivalent to p-values. Rather, FDRs give us a sense of what percentage of genes in the top set might be due to false positive by multiple testing of so many genes. Using FDR of $< 10\%$ is commonly accepted in expression profiling studies. However, relaxing FDR to something like 25% would still suggest that $\sim 75\%$ of genes in this expanded set would be true positives. Many nominally significant genes that miss a given FDR cut off may well be true positives but false negatives. Therefore, in the Supplementary Data files, we have provided p-values and estimated FDRs for all genes examined. Future studies profiling larger numbers of samples may uncover additional significant genes meeting the FDR threshold that were also nominally significant in our study.

In relation to the normality assumption, at what levels are the significantly associated genes expressed, compared to other genes? A preference for lowly expressed genes could point to a bias toward normality-violating genes.

Taking the set of 375 top genes with SSV-associated altered expression (using 1Mb region and distance metric method), we do find differences in the average gene expression between over- and under-expressed genes. However, this would not point to a bias towards normality-violating genes. For all ~18K genes profiled, the average log2 expression across the 854 tumors was 2.52. For the set of 324 over-expressed genes, the average log2 expression was 1.74, and for the set of 51 under-expressed genes, the average log2 expression was 3.29. The set of over-expressed genes was a bit lower expression on average, and the set of under-expressed genes was a bit higher expression on average. Notably, TERT, a gene associated with SSV-mediated up-regulation in many different studies and cancer types, has very low average log2 expression, at 0.22. But for the tumors harboring a nearby SSV breakpoint, the expression is high, as shown below:

We would conclude that, in many cases, the genomic rearrangements would alter the regulatory landscape in such a way that genes that normally have low expression in cells are strongly up-regulated. For other genes that normally have high expression in cells, SSV breakpoints would effectively knock the expression down. This pattern relates to the biology of the phenomenon we are examining, rather than issues regarding normality assumptions (which the log2 transform would alleviate).

Other comments:

Pg 5 row 103, Fig S1a: "patterns markedly differing from those of adult brain tumors from TCGA". The adult reference is just a single cancer type (glioma), so perhaps hard to make general conclusions? Some of the pediatric cancers may in fact have profiles that are similar to LGG.

Regarding Figure S1a and main Figure 2e, the point we wished to make is that we need to study pediatric brain cancer as a separate disease from adult brain cancer. For example, the CNA heat maps would show more overall differences than similarities between pediatric and adult tumors. We have known that different sets of genes are targeted in the setting of pediatric brain tumors compared to adult cancers, and our study reveals that different sets of genes would be SSV-altered as well. A general reader might otherwise assume that the previous TCGA studies of adult gliomas would be representative of pediatric brain tumors. However, there needed to be a study of SSVs in pediatric glioma separate from PCAWG or

other genomics studies of adult cancers. Future studies might also try to find commonalities between pediatric and adult brain cancer, for certain subtypes.

Fig. S1b: the large number of CNA amplifications associated with e.g. translocations seem surprising, as earlier studies have shown that a large fraction of them are specifically tandem duplications, as expected. Perhaps an overview of the absolute number of the different SVs types - or their relative abundances - could be useful. Translocations are often less common than e.g. tandem duplications, but S1b makes me think it is very different here? Some tandem duplications lead to gene fusions (e.g. KIAA1549-BRAF mentioned below) - I assume these are still classified as tandem duplications?

The breakdown of SSV breakpoints by type (shown below) is very similar, in relative distribution, as the SSV-gene associations in Figure S1b. The CBTC dataset distributions are also similar to what we observed previously for TCGA adult cancers (Zhang et al., Cell Reports 2018, PMID: 29996110). In that previous study, we also observed less representation of tandem duplications as compared to translocations (below, BND=translocation).

SSV type	Number breakpoints	% of total
BND	56231	34.8%
DEL	58990	36.5%
DUP	19972	12.3%
INV	25618	15.8%
INS	998	0.6%

We find that SSVs of all types—interchromosomal translocation, deletion, tandem duplication, insertions, and inversion—may be associated with gene-level copy gain. As a group, breakpoints of tandem duplication and inversion were significantly enriched for amplification events involving nearby genes. While SSVs may be balanced or unbalanced in terms of CNA within the immediate vicinity of the breakpoint (e.g., involving deletions, insertions, or tandem duplications), the associations in Figure S1b would involve a broader level of associations of CNA with SSVs, consistent with our previous observations in adult cancers.

In the revised Data file S4 (fusion calls), for the fusion calls with SSV breakpoints for one or both genes, we have added columns for the type of SSV. From this, we see that KIAA1549-BRAF fusion involves tandem duplications, as expected.

Fig. 1b: at least to me, this figure is difficult to understand and does not illustrate to points made in a very obvious way.

We intended this figure to illustrate how gene CNA events often occur in association with nearby SSV breakpoints. We had demonstrated this in our previous studies of adult cancers (e.g., Figure 1 of Zhang et al., Cell Reports 2018, PMID: 29996110). In the current manuscript, we tried to show this relationship in a straightforward way using Venn diagrams. The tumor-gene associations represented here (one circle for SSV breakpoints, one for either amplifications or deletions) were taken from the matrix of all gene X tumor sample pairings (~18K genes X 854 tumors). So, for a given gene and tumor pairing, we ask whether the gene had a nearby breakpoint for that tumor, and we also ask if the gene is amplified or

deleted in that tumor. The Venn diagrams show that the overlap between gene CNA events and breakpoints nearby genes is greater than chance expected, involving specific cytobands. In the manuscript revision, we added a bit more in terms of all gene X tumor pairing being considered in this analysis.

The text regarding fusions fails to even mention the extent to which fusions were in frame or not. If selection is generally absent, i.e. these are just random events or due to technical noise, about 1/3 should be in frame. In particular, it would make sense to mention whether the highlighted cases (e.g. BRAF) are in frame.

In the manuscript revision, we have added a new Supplementary Figure 4b. This figure shows the fraction of in-frame fusion predictions by Arriba or STAR-fusion algorithms, for all candidate fusion events, and for fusion events with SSV breakpoints being found within one or both genes, with or without a high expression association, as indicated. (Here we used the same fusion event categories as originally used in main Figure 4a). P-values are by one-sided Fisher's exact test. As we now note in the Results section, RNA-seq-based fusion predictions with SSV support were highly enriched for in-frame fusions, which would be in line with the reviewer's expectation. Of all the fusion candidate events we considered, around 5% were in-frame. However, of the candidate events with strong SSV and expression support, ~45% were in-frame.

Also, we have revised main Figure 4c to indicate which of the fusions represented involve in-frame fusion events. The new figure is shown above in response to a comment from reviewer #1. Most of the fusions in Figure 4c are in-frame, including all of the ones that were previously known.

To Data File S4, which annotates the RNA-seq-based fusion prediction according to support from SSV and differential expression, has been updated to indicate which predictions are in frame.

Is it BRAF-KIAA1549 or KIAA1549-BRAF?

We have come across both of these in the literature (perhaps a few more articles using the KIAA1549-BRAF term). Our previous convention was to list the two genes alphabetically. We have changed this in

the revision to “Gene1-Gene2”, as provided by the Arriba/STAR fusion prediction output files. This would represent the above fusion is question as KIAA1549-BRAF.

Fusion findings - and the same can be said generally for many findings in the study - would benefit from being presented while briefly referring to the existing literature. E.g. KIAA1549-BRAF is known and described since before, but the reader has no way of knowing whether this is a novel finding or not. Is the frequency consistent with earlier reports? This particular case arises due to a local tandem duplication. This is also quite essential information and rather different from e.g. an inter chromosomal translocation, but not mentioned. Naturally, there needs to be a balance due to limited space, but I believe some of the results need to be put into context a bit more with respect to prior findings.

The way we refer to the fusions previously identified is simply to put a literature reference next to the fusion name: “...fusions included...TRIM24-BRAF (n=3)[ref to article on this fusion], FGFR1-TACCI (n=3)[ref to article on this fusion], FGFR2-SHTN1 (n=2) [ref to article on this fusion], and MYB-QKI (n=2) [ref to article on this fusion].” For the fusions that have previously been found, our study would not necessarily be adding more information from what was previously determined. However, cataloging how SSVs are associated with fusions in the CBTTTC cohort would still be an important exercise.

In the revised Data file S4 (fusion calls), for the fusion calls with SSV breakpoints for one or both genes, we have added columns for the type of SSV. From this, we see that KIAA1549-BRAF fusion involves tandem duplications, as expected. (We link at most one SSV for each gene for each fusion event, so a few of the KIAA1549-BRAF entries may not have tandem duplication indicated, but the general pattern remains clear.)

Again in Fig. 4b, in the case of classical gene fusions it would be important to consider the reading frame (or clarify that consideration has been taken)

Most of the Figure 5b (formerly Figure 4b) fusion events are also represented in Figure 4c (formerly 3c), which now indicates which fusions represent in-frame fusions. Data file S4 also provides in-frame status for each fusion. The vast majority of the fusions represented in Figure 5b are in-frame. The BRAF fusions are the most represented by far, and all but 2 of the 120 BRAF fusion events represented involve in-frame fusions.

Fig 6, SVs vs survival: is the analysis performed on the complete cohort? Different cancer types represent completely different diseases and may thus have wildly different survival properties. In principle, results like this can be obtained due to a single cancer type having unusually high SV burden, and also short survival - which does not necessarily mean much biologically speaking. At least some more details are needed in the text such that the results can be properly interpreted. Also, references to the prior literature regarding SV or CNA burden vs survival would be useful such that it is more clear to what extents novel insights are being gained (again).

Figure 7a (formerly Figure 6a) does use all 759 patients, one tumor per patient (so, no counting of multiple tumors for the same patient in this case). P-values shown are by either univariate Cox or stratified Cox according to tumor type (i.e., the “corrected” p-value, p=0.002). The stratified Cox would account for the differences in patient survival between the different diseases represented. In the manuscript revision, we have made this clearer in the Figure 7a legend. In the revised manuscript, we

have noted previous studies making similar observations in adult tumors involving SV or CNA burden (Zhang et al. Genome Biology, PMID: 31610796; Hieronymus et al., eLife, 2018, PMID: 30178746).

Row 281: this another examples where a literature reference might be useful - is it known that high CNA or SV burden is associated with worse survival?

In the revised manuscript, we have noted previous studies making similar observations in adult tumors involving SV or CNA burden (Zhang et al. Genome Biology, PMID: 31610796; Hieronymus et al., eLife, 2018, PMID: 30178746).

SSV burden vs. gene expression: also here, a linear model can be problematic, and results from permutations (within cancer types) would be useful to assess the true FDR.

Under an earlier comment by the reviewer, we further address issues regarding our use of linear models, which follows the same approach we used previously for adult cancers (Zhang et al. Genome Biology, PMID: 31610796). Importantly, we log-transformed both SSV burden and gene expression values, making the data align better with the model assumptions. We did not use permutation testing to estimate the FDR, but rather we used the Storey and Tibshirani method, described above.

Row 325: was the mutual exclusivity with P53 significant (Fisher's exact test)?

It was not. It's generally difficult to establish significance in mutual exclusivity, as the events not involving the other gene are a large subset of the total.

Methods, Row 506: please explain the weighted method more clearly.

We have added some more detail on the weighted method, including the following: "The distance metric method provides a single result for each gene across the samples, representing genes consistently altered across the entire ± 1 Mb region examined. The 1Mb window was used as genome rearrangements may involve the translocation of enhancers, which may impact genes within a distance of ~ 1 Mb[ref]. The closer that SSV breakpoints are to the gene start in relation to altered expression, as compared to tumors without breakpoints, the more significant the association between expression and SSVs. Breakpoints further away from the given are less weight than breakpoints in closer proximity." We first described our method in Zhang et al. (Genome Biology, PMID: 31610796), which reference includes some diagrams to help illustrate the approach used.

Minor:

Although the manuscript is generally well written, I believe some more careful proofreading is needed to ensure all sentences are completely clear and grammatically correct. Examples: Grammar row 137: "These results lent support to our treating each tumor sample as a separate disease entity in the downstream analyses."

We have run our manuscript through Grammarly.com premium. This helped us tighten up some of the longer sentences.

Row 208, missing word.

Added. Thank you.

Reviewer #4 comments:

Somatic structural variants (SSVs) are an understudied class of variation in cancer, and their global impact on gene expression in many tumor types – including pediatric brain tumors is largely unknown. The authors combine cancer genome and RNA sequencing data from 854 tumors of thirty different childhood cancers to uncover associations between somatic SVs and transcriptomic changes. They describe different mechanisms leading to expression differences including gene fusions, altered cis-regulation events, gene disruption events, and altered gene dosage in this regard. The authors also associate high overall SSV burden with gene aberrations. From a methodological perspective, I did not notice any innovations in this study. The approaches described here have been used in prior applications of the same authors published in Nat Common and Cell Reports, which in many ways read quite similar to the current paper – with the exception that the focus of this manuscript is on childhood tumors which in prior studies relating SSVs to transcriptomic changes have received somewhat less attention. With respect to the choice of tumour types, I thus acknowledge that I do see a need for a study to search for tumor-driving SSVs, especially since this class of variation has received less attention in prior studies than copy-number changes or SNVs.

We thank the reviewer for evaluating our work.

While the study is largely carried out with previously devised methodology, I have a number of issues with the interpretation of the data. The authors seem to assume throughout the manuscript that correlation (association) can be equated with causality, which is worrisome. But clearly, just because two things (ie the presence of an SSV and a gene expression measurement) correlate does not necessarily mean that one causes the other (in this case that SSVs deregulate gene expression). Prior studies have used a number of ways to address the issue of causation vs correlation with respect to SSVs and gene expression dysregulation (including by searching for evidence of allele-specific expression, by investigating enhancer elements, or by performing 4C-seq based validations; see e.g. PMID: 27869826; 32632335, PMID: 28726821). I found the lack of appreciation of these important aspects in the current manuscript disappointing. If the authors get the chance to submit a revised manuscript, I would at least expect that they acknowledge this key issue, and demonstrate by using 4C-seq validations and/or haplotyping to generate allele-specific data further support/verification of their conclusions that genes are “deregulated by somatic rearrangements” (as currently stated e.g. in the title).

In the revised manuscript, we have added a new main figure, Figure 3. This figure explores SSVs associated with disruption of topologically associated domains (TADs) and translocated enhancers, as well as allele-specific expression patterns. These analyses would get at the issue of causation versus correlation, as raised by the reviewer. A sizable number of tumors underlying each SSV-gene association could be explained in terms of either TAD-disruption or enhancer hijacking, as detailed in the manuscript revision. Nevertheless, other mechanisms may be at work in SSV-mediated gene deregulation, not all of which may necessarily be known or observable. For example, in adult tumors, we previously found that rearrangement of genomic regions normally having higher or lower methylation is often involved in SSV-associated CGI methylation alterations and associated with altered gene expression (PMID:31610796). However, we do not have methylation data for the CBTC cohort, so we cannot catalog which SSV-gene associations involve altered methylation.

By design, our analytical approach does not assume the specific mechanism of altered expression. It is inherently challenging for us to state definitely the mechanism for a given sample with altered expression for a given gene. We might envision many plausible mechanisms. For example, a deletion SV may remove a small repressive regulatory element near the gene, causing the gene to be over-expressed; in another case, a tandem duplication SV may amplify the region surrounding a gene, but in the process breaks a promoter element, causing the gene itself to not be expressed; in other cases, there may be an SV breakpoint that is closest to the gene but which is having no effect, but then another SV right next to the first SV is actually the one that is driving the gene (e.g., a translocation SV bringing an enhancer element in proximity). The regulatory landscape for any given gene, in most cases, has not been fully elucidated. An analytical approach that assumes too much in terms of mechanistic effects is likely to be overly complicated and wrong in most instances. While for many SSV-expression associations, we may not identify the likely mechanism in terms of TAD disruption or enhancer hijacking, some other unbeknownst mechanism may still be involved. Therefore, such associations without an attributed mechanism may still be causal rather than correlative.

Below, we present our results exploring SSVs associated with disruption of TADs and allele-specific expression, and we present associations with enhancer hijacking under the next comment by the reviewer. Using data on TAD coordinates in human cells, we categorized all SVs in the CBTC dataset by those that were TAD disrupting (i.e., the breakpoints span two different TADs) versus those that were non-disrupting (i.e., both breakpoints fell within the same TAD). The results are shown in the plot below (new Figure 3c). For SSVs associated with gene over-expression, we observed an enrichment for TAD-disrupting SSVs (Figure 3b, $p < 1E-80$, chi-squared test), consistent with previous observations in adult tumors.

To provide a visualization of SSVs and TAD disruption involving specific genes, the new Figure 3f (also shown below) depicts the TERT locus (left) and MYB locus (right) and associated TADs and SSVs. The top portion represents TADs as Hi-C-based contact maps, with gray shading indicating locus interactions (darker, stronger interactions). The bottom portion represents gene expression levels of TERT and MYB corresponding to SSV breakpoints located in the genomic region. SSV breakpoints are annotated as TAD-preserving (i.e., both breakpoints fall within the same TAD) or TAD-disrupting. Dotted lines denote breakpoints within the same sample, and solid lines denote common SV event.

While the above patterns involving TAD-disruption are associations, globally, these are statistically significant. They would provide some context for a likely mechanism underlying the relationship between SSVs and deregulated expression (while other mechanisms would also be at work).

We also used cis-X (one of the papers noted by the above reviewer comment) to evaluate allele-specific expression for particular genes. The cis-X program is rather resource intensive, and so we focused here on TERT and MYB genes, genes which are provided as specific examples in our new TAD/enhancer figure (Figure 3). The new Figure 3b (also shown below) demonstrates that SSV-associated up-regulation of TERT and MYB appears allele-specific. Taking the set of tumors with SSV breakpoint within 100kb of TERT or MYB, respectively, we plot significance of allele-specific expression pattern (binomial test using cis-X) against total gene expression. Most of the tumors with both breakpoint and elevated expression had allele-specific expression. As might be expected, the greater the level of total expression, the higher the significance of allele-specific pattern (since more reads would be found covering the one allele).

Our knowledge on the location of cis-regulatory elements and enhancers along with genome of distinct tissues and cell types has vastly grown over the last decade. Can the authors make use of this resource of information, to evaluate whether in cases where intergenic SSVs correlate with gene expression this is associated with the specific juxtaposition (enrichment) of cis-regulatory elements / enhancers near those genes that become overexpressed?

In the revised manuscript, we have added a new main figure, Figure 3. This figure explores SSVs associated with disruption of topologically associated domains (TADs) and translocated enhancers. Below, we present our results exploring SSVs associated with enhancer hijacking.

We utilized the enhancer annotations as provided by Kumar et al. (Cell 2020, PMID: 32084333). We tabulated SSV breakpoint-to-gene associations involving enhancer translocation within 0.5Mb of the SSV breakpoint in proximity to the gene (assuming no other disruptions involving the region), where the unaltered gene either had no enhancer within 1Mb or had an enhancer further away from the gene than the translocated enhancer. We considered only SSVs with breakpoints on the distal side from the gene in this analysis. In other words, for genes on the negative strand, the upstream sequence of the breakpoint (denoted as positive orientation) should be fused relative to the breakpoint coordinates, and for genes on the positive strand, the downstream sequence of the breakpoint (denoted as negative orientation) should be fused relative to the breakpoint coordinates. Each enhancer search involved just one breakpoint and gene pairing, with only the SSV breakpoint closest to the start of each gene being considered for each tumor in the instance of multiple breakpoints being detected.

We found that SSV breakpoints involving gene over-expression were enriched ($p < 1E-8$, chi-squared test) for putative enhancer translocation events, with the rearrangement bringing an enhancer within 500kb of the gene. The new Figure 3d (also shown below) shows the percentages of SSV breakpoint associations involving the translocation of an enhancer, as tabulated for the entire set of SSV breakpoint associations with breakpoint mate on the distal side from the gene, as well as for the subsets of SSV breakpoint associations involving altered gene expression.

The new Figure 3e (also shown below) shows, by gene and by histologic type, the number of SV breakpoint associations involving the translocation of an enhancer, which involved 98 genes and 192 tumors. A new Excel tab in Supplementary Data File 2 provides information on the SSVs associated with enhancer hijacking and gene over-expression.

The above analysis has some limitations. As noted above, in the instance of multiple breakpoints being detected in association with a given gene in a given tumor, we considered only the SSV breakpoint closest to the gene's start. A possible scenario would be one where SV breakpoint that is closest to the gene has no effect, but then another SV right next to the first SV is the one that is driving the gene (e.g. a translocation SV bringing an enhancer element in proximity). At the same time, the enhancer hijacking associations are globally statistically significant. The results would provide some further context as to a likely mechanism underlying the relationship between SSVs and deregulated expression (while other mechanisms would also be at work).

The number of SSVs that the authors report as correlating with gene expressions seems (at least to me) unexpectedly high, which is worrisome. Could this be a result of the methodology used, which might be too permissive? The authors should provide QQ plots. What is the inflation factor lamda?

We do see a lot of genes associated with SSV breakpoints. This phenomenon would be entirely consistent with the results of our previous studies examining SSVs in adult cancers. Aspects of this phenomenon, as observed in both PCAWG and CBTC cohorts, include: hundreds of genes recurrently impacted, SSV breakpoints as far as 1Mb from the gene contributing to deregulation, rearrangements involving widespread CNA patterns, many more genes with increased over decreased expression associated with SSV breakpoints, and over-expressed and under-expressed genes respectively representing known oncogenes and tumor suppressor genes.

In the revised manuscript, we have included q-q plots for the SSV vs expression correlations. The results of this analysis are shown below and in the new Supplementary Figure 2b in the revised manuscript. The figure shows QQ plots of the linear regression p-values from the SSV events versus expression correlations (correcting for both cancer type and copy number), for SSV breakpoints falling within each of the indicated regions surrounding the gene.

The above results are consistent with widespread biology-driven associations between SSV event and expression, involving hundreds of genes, after correcting for expression patterns associated with tumor type or copy number. The p-values appear well-calibrated, and we see many more significant p-values over chance expected (indicated by the red line). In contrast, the bottom right QQ plot represents results from the SSV events being randomly permuted relative to the expression profiles (distance metric method using 1Mb region). Each plot includes a genomic inflation factor (lambda value). Though this measure would be taken from Genome-Wide Association Studies—and would seem more applicable for those types of studies rather than for differential expression analyses, we find the lambda values to be close to 1 for our study results. The plots show that there are many significant genes, such as what a typical SAM plot would show where widespread differential patterns above FDR are observable.

Our previous paper published with the PCAWG consortium (Zhang et al., Nature Communications 2020, PMID: 32024823) also features q-q plots in a supplementary figure. Interestingly, the PCAWG Steering committee members also asked to see q-q plots before we submitted that manuscript. The fact that so many genes are SSV-associated would be due more to biology than to methodology artifacts. If we were to correlate gene expression with CNA, we would similarly see widespread associations, more than our SSV associations. Other approaches, such as CESAM (Weischenfeldt et al. Nat Genet 2016, PMID:27869826), while similar to ours in its use of linear modeling, may also carry out independent filtering, e.g., limit SSVs of interest to those disrupting TAD boundaries. This filtering will result in fewer genes in the top results because the search space is more limited by definition. In contrast, our method does not assume mechanism of deregulation, so this would identify more genes with SSV-expression associations.

REVIEWER COMMENTS

Reviewer #2 (Remarks to the Author):

The authors have made an ambitious effort to revise the manuscript and - perhaps even more so - to respond very exhaustively to the questions. My main remaining issue relates to FDRs and permutations:

Although I want to avoid a prolonged back-and-forth discussion over multiple revisions, I have to say I don't find the answer completely satisfactory. About the linear model, there is no doubt that it does "a fairly good job of identifying significant patterns" and like the authors, I fully agree the findings "do not as a whole represent chance associations". There is also no confusion regarding FDRs being computed separately from the permutation testing. My points were simply about the risk of inflating the number of reported associations, which I think is very real. The model assumptions will be violated sometimes even following log transformation using this model (likely in a not negligible way), but that doesn't mean it's a bad idea and I agree that the approach has many virtues. Still, permutations are a very good reality check that helps bypass these problems - and in this case in fact your permutations suggest the Storey-Tibshirani-derived FDRs may be inflated. That "many permutations could result in a dataset that is not far removed from the original" is just another way of saying "the dataset is small and statistical power is therefore limited", a fact that has to be accepted. Yes, a parametric model followed by a theoretical FDR correction would yield more power - but it comes at a price (risk of exaggerating). "Less divergence from the actual data" is not a bad thing: the permutation strategy should relate directly to the main hypothesis being tested (associations between genomic alterations and RNA expression). Under the null hypothesis, permuting the expression data within tumors does not change the data from the point of view of your model - it is still the same, which is good. Permuting the whole cohort radically changes your data: the within-cancer type variance is now going to be completely different for many genes. So one has gone beyond scrambling the data such that the actual effect of interest is gone (i.e. trying to simulate the null hypothesis), to generating a dataset completely unlike the real tumor expression data, which makes it a bad point of reference for assessing your findings. One absolutely cannot limit the permutation analysis to genes being significant below a certain P-value in the original analysis, try e.g. doing some t-tests on random noise and apply the same strategy. I don't understand the "singular problems" argument - why don't you have these problems in the unpermuted data, and why are there more of these problems when permuting within each cancer type? To summarize: 1) it would be valuable to readers to, along side your theoretical FDRs, present the outcome of the permutations in a more prominent and quantitative way (main text and perhaps also main figure) and 2) the current permutation strategy is not ideal and may not provide a suitable background distribution in my opinion.

Reviewer #4 (Remarks to the Author):

The authors have been able to improve their manuscript in response to the comments I raised. I have no further comments.

Our responses and comments are marked in red italics.

Reviewer #1 comments:

Paraphrased by the editor (no remarks to the author provided): The reviewer is still concerned with the resource value provided by the manuscript as there are insufficient re-analyses using molecular classification to define the tumour types and the inclusion of the group of tumours classified as “Other”, which in the reviewer’s opinion should be omitted.

As requested by the editor, we have provided a detailed discussion in the revised manuscript on the choice of histological classification rather than molecular subtypes (taking points from our previous rebuttal). From Results, page 10 of the manuscript revision: “Using molecular subtype instead of histologic type in the SSV-expression modeling, or including only tumors originating from brain-specific cell types in the analysis, yielded very similar SSV-gene associations uncovered above using the full cohort of 854 tumors (Supplementary Figure S3). Based on unsupervised clustering of RNA-seq data, tumors broadly segregated according to histologic type (Supplementary Figure S3a). Our statistical models included histologic type of the tumor as one of the covariates when defining genes altered in association with nearby SSV breakpoints. Still, the histologic type covariate did not heavily influence the top results (Supplementary Figure S3b). While many histologic types may be comprised of molecular subtypes, in a global molecular analysis, the most-represented histologic types—including PLGG, PHGG, MBL, EPMT, and ATRT—all formed fairly homogenous groups (Supplementary Figure S3a). Consistent with the approach of other studies of SSVs and expression [refs, including studies outside of our group], our pan-cancer analyses used histological classification rather than molecular subtype. Furthermore, the CBTC’s regulatory for CNS tumors intentionally allowed for the broad collection of abnormal cell growth, not necessarily specific to brain-specific cell types. Likewise, all data from 854 tumors that were available from CBTC formed the basis of our study. However, in an alternate analysis, we first removed tumor types that originated from cell types not specific to the brain, along with metastatic secondary tumors and tumors that did not fit within the more common CBTC tumor type designations. The remaining set of 793 tumors yielded very similar SSV-expression associations to those of the 854-tumor set (Supplementary Figure S3c and Data File S2).”

As indicated above, we have added the gene-level SSV-expression associations for the 793-tumor set to Data File S2, for any readers who may wish to examine those.

Reviewer #2 (Remarks to the Author):

The authors have made an ambitious effort to revise the manuscript and - perhaps even more so - to respond very exhaustively to the questions.

We thank the reviewer for evaluating our work.

My main remaining issue relates to FDRs and permutations:

Although I want to avoid a prolonged back-and-forth discussion over multiple revisions, I have to say I don’t find the answer completely satisfactory. About the linear model, there is no doubt that it does “a fairly good job of identifying significant patterns” and like the authors, I fully agree the findings “do not as a whole represent chance associations”. There is also no confusion regarding FDRs being computed

separately from the permutation testing. My points were simply about the risk of inflating the number of reported associations, which I think is very real. The model assumptions will be violated sometimes even following log transformation using this model (likely in a not negligible way), but that doesn't mean it's a bad idea and I agree that the approach has many virtues. Still, permutations are a very good reality check that helps bypass these problems - and in this case in fact your permutations suggest the Storey-Tibshirani-derived FDRs may be inflated. That "many permutations could result in a dataset that is not far removed from the original" is just another way of saying "the dataset is small and statistical power is therefore limited", a fact that has to be accepted. Yes, a parametric model followed by a theoretical FDR correction would yield more power - but it comes at a price (risk of exaggerating). "Less divergence from the actual data" is not a bad thing: the permutation strategy should relate directly to the main hypothesis being tested (associations between genomic alterations and RNA expression). Under the null hypothesis, permuting the expression data within tumors does not change the data from the point of view of your model - it is still the same, which is good. Permuting the whole cohort radically changes your data: the within-cancer type variance is now going to be completely different for many genes. So one has gone beyond scrambling the data such that the actual effect of interest is gone (i.e. trying to simulate the null hypothesis), to generating a dataset completely unlike the real tumor expression data, which makes it a bad point of reference for assessing your findings. One absolutely cannot limit the permutation analysis to genes being significant below a certain P-value in the original analysis, try e.g. doing some t-tests on random noise and apply the same strategy. I don't understand the "singular problems" argument - why don't you have these problems in the unpermuted data, and why are there more of these problems when permuting within each cancer type?

We value these comments, and we especially appreciate the reviewer's summary below, which provides us a specific path to address these concerns. We will focus our response on the summary statement below, which response will touch on the above.

To summarize: 1) it would be valuable to readers to, along side your theoretical FDRs, present the outcome of the permutations in a more prominent and quantitative way (main text and perhaps also main figure) and 2) the current permutation strategy is not ideal and may not provide a suitable background distribution in my opinion.

To directly address the reviewer's two summary points, we have revised the manuscript as follows. Supplementary figure 2a (also shown below) now presents FDR estimation results from permuting within tumor types (per the Reviewer's summary point #2). The reviewer's comments regarding our "singular" issue rang true, and so we went back to our code and corrected the problem, where now the within-tumor-type permutations would involve all genes being tested.

(As we had expected, the within tumor type permutation did lead to a somewhat increased estimated FDR, as there are fewer combinatorial possibilities here. We should note that it is the SSV profiles that are being shuffled in the permutation testing, while the expression and tumor type relationships remain intact. SSVs and associated CNAs are sparse in pediatric brain cancer, and do not as strongly associate with specific tumor types, e.g., Supplementary Figure 1a).

Supplementary figure 2a (shown above) presents estimated FDRs involving the top significant genes with SSV-expression associations ($p < 0.001$, distance metric method, correcting for both tumor type and CNA). FDRs are estimated using the following methods: 1) Storey and Tibshirani method, 2) permutation of SSV profiles across all 854 samples with respect to the expression profiles, 3) permutation of SSV profiles with respect to the expression profiles but shuffling within tumor type (given the strong expression versus tumor type associations), 4) permutation of SSV profiles with respect to the expression profiles but shuffling within tumor type and using a large simulated dataset. All permutation testing results are based on 1000 permutations. For the large simulated dataset, five copies of the actual 854-tumor matrices were concatenated together to make one large dataset of 4270 samples, where the same relationships as represented in the 854-tumor dataset are present, but with the larger sample size allowing for more permutations that have little or no overlap with the actual dataset (being closer to truly random distributions). As expected, the large simulated dataset shows fewer estimated false positive genes than permutation results using the original 854-tumor dataset, where a larger sample size affords more power. The results using the large simulated dataset attempts to get around the issue of the inherent limitation of the 854-tumor sample size as regards permutation testing. Permuting across all tumor profiles using the large simulated dataset (result not plotted) indicates that ~43 of the 295 $p < 0.001$ genes might be false positives by this estimate.

In addition, Supplementary Data File S2 provides an Excel tab with the top genes with $FDR < 10\%$ (1Mb region window, correcting for CNA, with FDR by Storey and Tibshirani method), with the corresponding gene-level FDR estimations by permutation testing (including within tumor type permutations) also being provided in the revised file. In picking out genes of potential interest using this file, readers can take the permutation results into account, as well as which genes represent known cancer genes or show up independently in the PCAWG/TCGA datasets (results also provided in the same Excel tab), along with any domain-specific knowledge on the part of the reader.

In the revised main text (page 8), we directly address how the estimated Storey and Tibshirani estimates compare with the permutation testing results of Supplementary Figure 2a and Data File S2 (per the Reviewer's summary point #1). The additional text in Results reads as follows: "In addition to results using statistical modeling and the Storey and Tibshirani FDR estimation method, permutation testing results reflected widespread significant patterns associating SSVs with expression (Supplementary Figures 2a and 2b). By permuting the 854 SSV breakpoint profiles within tumor types 1000 times, the 324 genes with Storey and Tibshirani FDR<10% (using the 1Mb window) would have an estimated FDR of 33% by permutation method (Data File S2), indicating that on the order of 67% of the genes would represent true positives. However, the number of tumor samples in the dataset limited permutation testing power, as some permutations would not be too far removed from the actual dataset. Permutation testing results using a large simulated dataset of over 4000 tumors yielded even fewer estimated false positives from the 854-tumor permutation results (Supplementary Figure 2a), with ~74% of the 324 genes presumably representing true positives (Data File S2). With very large sample sizes, datasets would be expected to yield permutation testing results much closer to the Storey and Tibshirani FDR estimates. In our downstream analyses below, we focused primarily on known cancer genes (including known fusions) and global associations involving the 324 genes."

The reviewer would understand that the CBTC datasets have relatively small sample numbers and that the statistical power for permutation testing is limited. The inherent limitations here lie not just with the dataset but also with non-parametric methods in general. In practice, non-parametric methods need lots more data than parametric methods because the former cannot rely on model assumptions. For a typical molecular biology experiment with ~4 samples per group, permutation testing offers absolutely no power (e.g., there are not enough possible permutations to establish $p < 0.05$). This limitation is why linear models have been so popular, and why the Storey and Tibshirani paper, for example, is heavily cited (close to 9000 times since 2003). With the 854-sample dataset, we can clearly show by permutation testing that the widespread phenomenon of SSVs associated with altered expression is nonrandom. One way to get around the sample number limitations is by using a simulated dataset, where we took five copies of the actual 854-tumor matrices and concatenated them together to make one large dataset of 4270 samples, where the same relationships as represented in the 854-tumor dataset are present in the simulated dataset. With the simulated dataset, we get FDR estimates closer to the Storey and Tibshirani approach because the larger dataset has more combinatorial possibilities to capture randomness better. While results using the large 4270 tumor dataset are based on simulated data, permutation testing itself represents a simulation. We can project that an extremely large dataset would bring us ever closer to a truly null distribution, which would likewise bring the permutation results even closer in line with Storey and Tibshirani estimates.

In the manuscript, the downstream analyses of Figures 3-5 focus primarily on the 324 genes with FDR<10% by Storey and Tibshirani. Our reading of the reviewer's summary comments is that the above would be acceptable. The Storey and Tibshirani estimate is that on the order of 292 of the 324 genes represent true positives. The permutation within tumor type estimate is that ~217 of the 324 genes should represent true positives. Permutation testing within tumor type, using the simulated large dataset, estimates that ~239 of the 324 genes represent true positives. Permutation testing across all tumors, using the simulated large dataset, estimates that ~267 of the 324 genes (82%) represent true positives. We expect an even larger simulated dataset would bring us closer to the Storey and Tibshirani estimate. Each of the above approaches would have its limitations. In many studies (e.g., those using GSEA), a q-value of

0.25 (in line with the above permutation results) would be considered acceptable, where an FDR is not equivalent to a significance p-value. Selection cutoffs must balance false positives with false negatives. In our results, many important pediatric cancer-associated genes have FDR<10% by Storey and Tibshirani method but FDR>10% by permutation testing, including known fusion-related genes such as KIAA1549 (part of the KIAA1549-BRAF fusion) and known cancer-genes TERT, CDK4, CDKN2A, NF1, which genes also show up in the PCAWG/TCGA analysis results. To not include KIAA1549-BRAF in the fusions figure 4 or the other genes highlighted in the pathways figure 5 would raise serious questions from disease experts. We also examine the 324 gene set for global associations (e.g., enhancers), where the true positives would be primarily driving the enrichment patterns. The FDR issue would be more of a concern if we were to highlight a specific and previously understudied gene as representing a core finding of our study, solely based on a nominal p-value or FDR value. However, this is something we are careful not to do.

Reviewer #4 (Remarks to the Author):

The authors have been able to improve their manuscript in response to the comments I raised. I have no further comments.

We thank the reviewer for evaluating our work.

REVIEWERS' COMMENTS

Reviewer #2 (Remarks to the Author):

The authors have taken the comments regarding FDRs seriously and have made appropriate changes to the paper. I am sure this study will be appreciated by many.

REVIEWERS' COMMENTS

Reviewer #2 (Remarks to the Author):

The authors have taken the comments regarding FDRs seriously and have made appropriate changes to the paper. I am sure this study will be appreciated by many.

We thank the reviewer for evaluating our work.